



# Analysis of high gas concentration and flux measurements at Swiss Beromünster tall tower

Andreas Plach[1,2], Rolf Rüfenacht[3], Simone Kotthaus[4], and Markus Leuenberger[2,5]

[1]Department of Meteorology and Geophysics, University of Vienna, Josef-Holaubek-Platz 2, 1090 Vienna, Austria
[2]Physics Institute, Climate and Environmental Physics, University of Bern, Sidlerstrasse 5, 3012 Bern, Switzerland
[3]Federal Office for Meteorology and Climate MeteoSwiss, Payerne, Switzerland
[4]Institut Pierre Simon Laplace (IPSL), CNRS, Ecole Polytechnique, Institut Polytechnique de Paris, 91128 Palaiseau Cedex, France
[5]Oeschger Centre for Climate Change Research, Hochschulstrasse 4, 3012 Bern, Switzerland

**Correspondence:** Andreas Plach (andreas.plach@univie.ac.at)

**Abstract.** The main research goal of this paper is to identify the most likely source regions of pollution events at Beromünster, Switzerland. Trace gas concentration and flux observations are essential to investigate emission source regions of greenhouse gases (GHGs) and other pollutants. Here we present an analysis of observations taken at an altitude of 212 m above ground at the Beromünster tall tower (2017 - 2020). The relatively high observation height — especially for flux measurements —

results in a large tower footprint, i.e., area sampled by the tower, predestined for a source analysis on a large scale. We identify high observations and analyze subsets of the concentrations and fluxes by atmospheric stability conditions (local vertical temperature gradient and atmospheric boundary layer height) to distinguish shorter from longer transport distances. And we split the data by prevailing wind direction as an additional separation of potential source regions. Furthermore, we perform inter-species correlation analyses and a parameterized Flux Footprint (FFP) estimation for the tower. We find that

pollution events at Beromünster tower are not associated with local emission sources, but are transported from further away — most prominently from the Northeast (NE) and Southeast (SE), where Zurich and Luzern are located, respectively. All species concentrations are highly correlated during winter, while in summer only a limited inter-species correlation exists. The parameterized annual mean FFP shows an extent of roughly 50-x-25 km along and perpendicular to the main wind axis, respectively.

## 1  Introduction

Continuous time series of atmospheric trace gas concentration and flux measurements are key to understand diurnal and seasonal variations as well as source regions of greenhouse gases (GHGs) and atmospheric pollutants. In this study, we analyze observations at a tall tower site close to Beromünster, Switzerland, to improve our knowledge of spatial and temporal variability of emission sources in this highly populated northern part of Switzerland (Swiss Plateau). For this purpose we use highly

accurate atmospheric $CO_2$, $CH_4$, $CO$, and $H_2O$ concentrations (Cavity Ring-Down Spectroscopy (CRDS); Crosson, 2008) and



$CO_2/H_2O$ flux measurements taken at Beromünster tower (flux measurements added recently; detailed technical description; Herrmann, 2019).

At the Beromünster tall tower, concentration measurements are taken at five altitude levels sequentially with additional flux measurements at the highest level (212.5 m above ground level; agl). Other European tall flux towers are located in Cabauw, NLD; Hegyhtsal, HUN; Norunda, SWE (Vermeulen et al., 2008). More tall tower sites have been established in recent years, for example within the Integrated Carbon Observation System (ICOS; e.g., Heiskanen et al., 2022). The measurement of $CO_2$ and $H_2O$ fluxes at Beromünster started in April 2016 and is based on the well-established eddy covariance (EC) technique (Aubinet et al., 2012; Mauder et al., 2021). Typically EC-fluxes are measured at so-called flux towers with observation heights of a few to 10s of meters, measuring fluxes for a very limited and local emission source/sink region, i.e., only a small area surrounding the tower is sampled for direct EC-fluxes (e.g., Schmutz et al., 2016). For Beromünster tower we estimate this footprint to be much larger, i.e., 10s of kilometers in its alongwind extent (see Sec. 3.3).

Several previous studies have investigated concentration measurements at Beromünster: Oney et al. (2015) gave an overview of the CarboCount CH network, which includes Beromünster. They found an average diurnal $CO_2$ variability of $-4$ to $+4$ ppmv at Beromünster and used the Lagrangian atmospheric transport model FLEXPART (Pisso et al., 2019; Stohl et al., 2005) to backtrace virtual particles from the tower to potential emission source regions affecting the observed concentrations, thereafter termed Concentration Footprint (CFP). The $CO_2$ variations were simulated to originate from the entire Swiss Plateau, i.e., 50 % of the particles were backtraced to be influenced by the surface within a distance of 130 - 260 km. Berhanu et al. (2016) and Berhanu et al. (2019) focused on a technical description of the observation system (CO, $CO_2$, $CH_4$, and $H_2O$ concentrations initially; $O_2$ added later). Satar et al. (2016) provided an analysis of seasonal and diurnal variations, inter-species correlations, and vertical storage fluxes along the tower for the first two years of data. They found atmospheric growth rates of 1.78 ppm yr$^{-1}$, 9.66 ppb yr$^{-1}$, and $-1.27$ ppb yr$^{-1}$ for $CO_2$, $CH_4$, and CO (for 2013 - 2014), respectively, and a very strong correlation of CO and $CO_2$ in winter ($R^2 > 0.75$), while no correlation in summer. Oney et al. (2017) discussed a method to determine the regional biogenic $CO_2$ signal based on co-located CO and $CO_2$ measurements, while Berhanu et al. (2017) provided a method based on $^{14}C$ measurements to identify the fossil fuel $CO_2$ component, the biogenic part was therein derived as the difference between total and biogenic $CO_2$. Furthermore, Bamberger et al. (2017) performed a comparison of the Beromünster tall tower site to a nearby mountain-top site (Früebüel; $\sim$ 30 km distance) with a similar observation height of roughly 1000 m asl. They found that the mountain-top site is more influenced by local wind systems and local emission sources than Beromünster, and that for $CH_4$ both sites could be used alternative to each other (e.g., for inverse modeling), if local $CH_4$ sources are well known, and can therefore be removed via filtering, while for $CO_2$ a substitution of one station with the other is more challenging due to differences in surrounding vegetation and soils causing a more pronounced diurnal cycle at the mountain top site. Finally, Rust et al. (2022) performed a top-down assessment (Bayesian inversion) of recent Swiss halocarbon emissions based on a measurement campaign at Beromünster.

Albrecht et al. (2012) provide a high resolution (500 m) anthropogenic $CO_2$ emission inventory for Switzerland (excl. natural processes). The main $CO_2$ emitters were found to be located in the urban areas and along major roads, most prominently in the vicinity of Beromünster: Zürich in the NE (20-30 km), Luzern in the SE (15-20 km), Bern in SW (60-70 km), and several



smaller cities in the NW and NE (20-60 km). The provided temporal $CO_2$ emission variability shows (1) a daytime emission peak for most included emission categories (e.g., energy / transformation, agriculture, and manufactures) or two peaks during morning and evening (road transport, non-industrial), (2) less emissions during the weekends, and (3) a seasonal maximum during winter (DJF) for all covered emission categories, except for agriculture, for which most $CO_2$ emissions occur in the months August to November. However, these temporal and spatial $CO_2$ emission patterns are complicated further by an overlay with natural processes, e.g, photosynthesis and respiration.

Hiller et al. (2014) discuss a high resolution (500 m) $CH_4$ emission inventory for Switzerland (anthropogenic plus natural) which was validated via a regional-scale atmospheric inversion from measurements including also observations from the Beromünster site in Henne et al. (2016). Most of the $CH_4$ emissions were located in the Northern part of Switzerland, on the Swiss Plateau, with especially high $CH_4$ emissions around the Beromünster site, and in NE-Switzerland. While there is no temporal variability included in the initial $CH_4$ emission inventory (Hiller et al., 2014), Henne et al. (2016) find a strong wintertime reduction of $CH_4$ emissions in agricultural areas, e.g., the area around the Beromünster site.

Since considerable diurnal and seasonal natural variations persist in the observed trace gas concentrations (shown in previously listed Beromünster-focused studies), we remove these temporal variations within our statistical method to be able to identify pollution events, and in a further step try to locate most likely source regions. We remove the natural variations by calculating diurnal-seasonal $90^{th}$ percentiles as references and define high trace gas concentrations/fluxes as the observations above these reference values (thereafter referred to as high concentrations/fluxes). Furthermore, we use the prevailing wind direction and atmospheric stability information — Atmospheric Boundary Layer (ABL) height and local vertical potential temperature gradient — to distinguish between cardinal direction and distance of the emission source regions. We perform this analysis for $CO_2$, $CH_4$, CO, and $H_2O$ concentrations, and as a novelty for the Beromünster EC-flux measurements.

Our period of interest starts with the availability of the ABL data used and covers the years 2017 to 2020. We focus only on times with valid trace gas concentration and at the same time valid EC-flux measurements (larger gaps Jan-Aug 2018 and May-Sep 2019; overview of data availability in Fig. S1). Furthermore, we use only observations from the highest level (212 m agl) since the EC-fluxes are exclusively measured at this level of Beromünster tower. Using the highest level observations also insures that we investigate the largest possible source region extent around the tower.

Additionally, we perform inter-species correlation analyses for $CO_2$, $CH_4$, CO concentrations, and $CO_2$ EC-fluxes to get indications of linked source processes. We conclude with a presentation of parameterized Flux Footprints (FFPs) estimates to illustrate the area from which direct turbulent EC-fluxes can be backtraced from the tower, as a comparison to previously published Concentration Footprints (CFPs; emission sensitivities maps; Oney et al., 2015; Rust et al., 2022), which show areas potentially influencing the observed concentrations via advective transport (see Sec. 2.4 and Fig. 1b for the difference between CFPs and FFPs). Following this introduction we describe the methods used (Sec. 2), discuss our results (Sec. 3), and give our main conclusions and an outlook (Sec. 4).





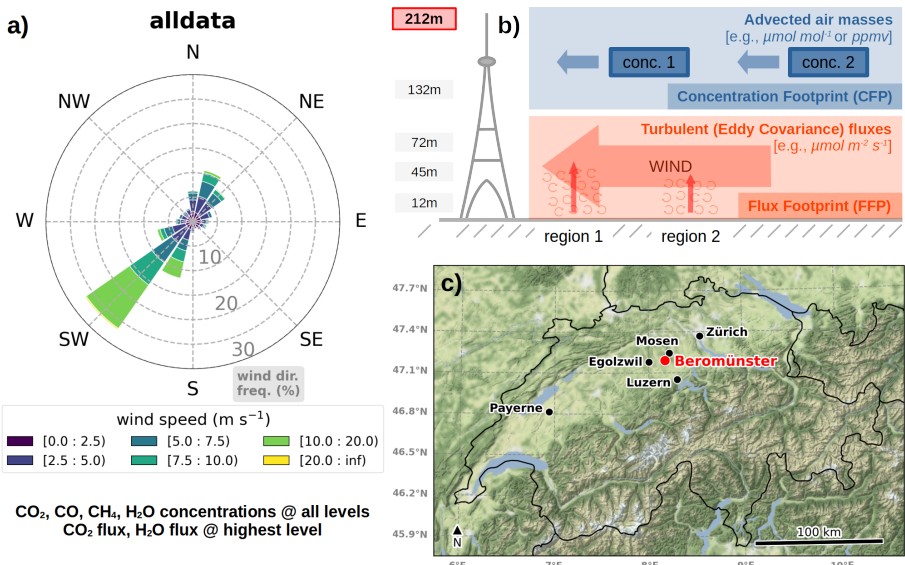

**Figure 1.** (a) Annual mean wind rose of the Beromünster site (entire time series; no data excluded; see Sec. 2.3). (b) Sketch of the Beromünster tower (47.190°N, 8.176°E, 797 m asl) with the five measurement heights indicated and sketch of advected air masses (with varying species concentrations) and turbulent (eddy covariance; EC)-fluxes, which reach the observation site directly from the respective sources in the immediate vicinity. (c) Overview map with discussed sites indicated. Map tiles by Stamen Design under CC BY 3.0. Data by © Open-StreetMap under ODbL.

## 2 Methods

### 2.1 General description of the Beromünster site

90 The Beromünster tall tower is an old radio tower converted to an atmospheric measurement site approximately 30 km southwest of Zurich on the Swiss Plateau, a landscape with relative smooth topography. The tower itself is located on a gentle hill top at an altitude of 798 m above sea level (asl), roughly 300 m above the surrounding valleys. Atmospheric concentration measurements are taken sequentially at five altitude levels — 12, 45, 72, 132, and 212 m above ground level (agl) and additionally EC-flux measurements are taken at the highest level (Fig. 1b). In this study we focus on concentration and flux measurements

95 from the 212 m agl level. The main wind axis is oriented in a northeast-southwest (NE-SW) direction (Fig. 1a; initially shown in Oney et al., 2015, updated in this study).

### 2.2 Data overview

In this study, we use several types of measurements. While (1) atmospheric trace gas concentrations and (2) eddy covariance (EC) fluxes are the main objects of our analysis, we additionally use (3) Atmospheric Boundary Layer (ABL) heights in

100 combination with vertical profiles of potential temperature observed at the tower to both evaluate the atmospheric stability and





as an input for Flux Footprints (FFPs), (4) global radiation measurements for a flux gap-filling procedure, and (5) additional meteorological data, including wind speed/direction, air temperature/pressure, and humidity at any inlet height of the tower:

1. Atmospheric $CO_2$, $CH_4$, CO, and $H_2O$ concentrations (Picarro Cavity Ring-Down Spectroscopy (CRDS) measurements)

2. EC-fluxes of $CO_2$ and $H_2O$ (LiCOR Li-7000 gas concentration analyser and a Gill WindMaster 1590-PK-020 anemometer)

3. ABL heights (derived from Automatic lidars and ceilometers (ALC) profile measurements (Lufft CHM15k) at Payerne using the STRATfinder algorithm; Kotthaus et al., 2020)

4. Global radiation measurements (average of two MeteoSwiss stations in close vicinity of the Beromünster tower)

The continuous measurements of $CO_2$, $CH_4$, CO and $H_2O$ concentrations at Beromünster were initiated in October 2012 and extended by $CO_2$ and $H_2O$ EC-fluxes in April 2016. Detailed descriptions of the technical setup, data processing concept, and calibration strategy of the concentration, and EC-flux measurements are available in Berhanu et al. (2016) and Herrmann (2019)respectively. Therefore only a brief summary is provided here. The availability of the datasets is summarized in Fig. S2.

### 2.2.1 Atmospheric concentration measurements and concentration footprints

The mixing ratios of $CO_2$, $CH_4$, CO, and $H_2O$ ($\mu$mol mol$^{-1}$ or ppmv) are measured with a Picarro Cavity Ring-Down Spectroscopy (CRDS; Crosson, 2008) analyzer (G2401) employing sequentially retrieved air samples from five different elevations (Fig. 1b). The ambient air from each level is measured for three minutes while only the last 60 seconds are used to avoid contributions from the previous level. Standard gases are measured on a weekly basis to calibrate the air measurements. Additionally, a working standard (6-hourly) and a target gas (daily) are assessed to monitor the measurement drift and the accuracy/long-term stability of the system, respectively. The long-term reproducibility and calculated accuracy are below the WMO target values (WMO, 2020, expert group recommendations therein).

Lagrangian transport models, such as FLEXPART (Pisso et al., 2019; Stohl et al., 2005), are often used to simulate so-called Concentrations Footprints (CFPs), also known as, emission sensitivity maps, by releasing virtual particles at the observation site and following them backward in time (e.g., for 4 days) to identify regions where and for how long these particles are close to the surface (e.g., within 100 m agl) and therefore susceptible to potential (surface) emissions. These CFPs can be used to explain hourly to daily variations in the concentration observations. Such a simulation was performed for Beromünster by Oney et al. (2015). In combination with an accurate emission inventory map, these CFPs can be used to reproduce observed concentration time series at the site.

### 2.2.2 Eddy covariance (EC) measurements

Another set of temporal highly resolved $CO_2$ and $H_2O$ concentrations are measured (at the highest level and with 20 Hz) with a LiCOR equipment (Li-7000, a near infrared non-dispersive laser spectrometer). Variations of $CO_2$ and $H_2O$ are used





in combination with the simultaneously measured wind components to derive $CO_2$ and $H_2O$ EC-fluxes (30-min integrated; $\mu mol\ m^2\ s^{-1}$). For the EC technique the absolute concentrations are less important, short-time concentrations variations are essential. Positive EC-fluxes arrive directly at the observation site from (surface) sources in the immediate vicinity and are responsible for short-time variations of observed concentrations. Correspondingly, negative EC-fluxes are directed towards sinks in the vicinity (Fig. 1b).


We calculate the EC-fluxes in several steps: In a first step, the LiCOR measurements — $CO_2$ and $H_2O$ concentrations, x-, y-, and z-wind components, sonic temperature (stemp), and speed of sound (sos) — are filtered for (1) faulty flow rates (rate by which the ambient air is sampled), (2) hard thresholds excluding unrealistically low/high values, and (3) a monthly 3-standard-deviation filter. In these steps up to 4 % of monthly data points are removed.


Then we process the filtered Licor data with the EddyPro Software (v7.0.6) to derive 30-min $CO_2$ and $H_2O$ EC-flux estimates followed by a post-EddyPro processing step in which we filter the EC-flux estimates — $CO_2$, $H_2O$, sensible, and latent heat flux: (1) Hard thresholds removing $\sim 20$ % of the data. (2) Removing data flagged with a bad EddyPro quality flag (removing $\sim 20$-25 %). (3) Applying a 3-std filter (removing $\sim 1$-2 %). All %-numbers are relative to the EddyPro processed data.

The last step in the flux processing is a gap-filling procedure (software package REddyProc v1.2; Wutzler et al., 2018),

basically filling gaps with fluxes of corresponding months, time of day, and similar global radiation regimes (Sec. 2.2.4). A more detailed description of the EC-flux measurements at Beromünster is available in Herrmann (2019).

### 2.2.3 Atmospheric Boundary Layer (ABL) measurements

Automatic lidars and ceilometers (ALC) are compact aerosol backscatter lidars that can be operated continuously with very low maintenance. They have a lower pulse energy compared to research-grade lidars which reduces their signal-to-noise ratio in the

upper troposphere but in the near range again they have a better coverage because of a shorter incomplete optical overlap. From the recorded vertical profiles of attenuated backscatter the height of the atmospheric boundary layer (ABL) can be retrieved (Kotthaus et al., 2022). In this study, we use ALC data from the MeteoSwiss site at Payerne (46.814°N, 6.943°E, 490 m asl), 100 km to the Southwest on the Swiss plateau.

Despite the distance it shall be noted that the two sites have comparable climates and ABL characteristics. The Lufft

CHM15k operating at 1064 nm is considered a high-power ALC so that ABL heights can be derived using the automatic STRATfinder algorithm (Kotthaus et al., 2020). The CHM15k measurements have increased uncertainties below a range of $\sim 250$ m, due to their optical design (bi-axial emitter/receiver) and consequent incomplete optical overlap. This means details of shallow layers can not be assessed by this data. Still, for the purpose of this study, where the ABL height (ABLH) data are mainly exploited to assess whether the layer height is above 500 m asl, the product has suitable quality.

### 2.2.4 Global radiation

Global radiation data is needed for the EC-flux gap-filling procedure (Sec. 2.2.2). We use a 30-min average of originally 10-min measurements at two MeteoSwiss stations in close vicinity to the Beromünster tower — Egolzwil (47.179°N, 8.005°E 522 m asl) and Mosen (47.244°N, 8.233°E 453 m asl) (Fig. 1c). Observations directly at the tower site are only available since





April 2017 at a 1-hourly resolution. The 1-hourly mean of the used Egolzwil/Mosen average are very consistent with the
1-hourly measurements at the tower (not shown; see Fig. S2).

## 2.3    Analysis of data subsets for high observations

The main goal of our analysis is to recognize high concentration and flux observations (Sec. 2.3.1) of the various species and
identify situations in which the percentage of these high observations is largest. Therefore, we analyze our dataset separated in
various subsets to account for different atmospheric stability (which we associate with shorter/longer transport; Sec. 2.3.2) as
well as the prevailing wind direction representing different cardinal direction of source regions (Sec. 2.3.3).

### 2.3.1    Definition of high concentration/flux observations

In a first step, we group our observations by month and time of day (2-hourly) and calculate median, MAD (median ab-
solute deviation), and $90^{th}$ percentile for each of these diurnal-seasonal subgroups (Fig. 2). This results in 12 x 12 me-
dian/MAD/percentile values per year (Fig. 2) — grouping applied to all data points; no separation by atmospheric stability
or wind direction yet (Fig. 3; Sec. 2.3.2 and 2.3.3). We performed tests with different definitions of "high observations", i.e.,
larger than mean plus two STDs (standard deviations) and larger than median plus two MADs (not shown). The patterns are
the same for all definitions, however, the $90^{th}$ percentile has the advantage of being independent of how the observations are
distributed.

These diurnal-seasonal statistical values are used to flag observations as high if they are above the respective subgroup $90^{th}$
percentile. This means each observations is evaluated with the respective diurnal-seasonal subgroup statistics. Furthermore, we
normalize deviations from the respective subgroup median by dividing the difference between the individual observation and
the corresponding median by the respective MAD.

In the next step we analyze subsets of the entire dataset, e.g., separated by prevailing wind direction, and investigate the
percentages of observations flagged as high in these "wind direction subsets". A respective subset is more prone to high
observations if the percentage of observations flagged as high is greater than 10 % — this is an important concept of the
analysis here.

We design our statistical method in this way to predominately remove natural variability caused by photosynthesis and
respiration (seasonal and diurnal) — $CO_2$ concentrations are in general lower in summer (photosynthesis dominates over
respiration) and higher in winter (respiration dominates), however also anthropogenic variability will be removed to some
degree (e.g., weekday/weekend variations of traffic induced anthropogenic $CO_2$ emissions). The same concentrations may
therefore be flagged as high in summer, while in winter it may be below the $90^{th}$ percentile. Ideally exactly 10 % of data
points should be above the $90^{th}$ percentile if we look at subsets of the entire dataset, e.g., winter (*DJF*), summer (*JJA*), spring
(*MAM*), and fall (*SON*) in Fig. A1. However, due to the partly small sample size in the diurnal-seasonal subgroups (when
calculating the percentiles) the percentages deviate slightly from 10 % — exactly 10 % would only be reached with a sufficient
sample size. Due to this removal of the diurnal-seasonal variability, positive deviations from 10 % should mainly be caused by
(anthropogenic) pollution events.



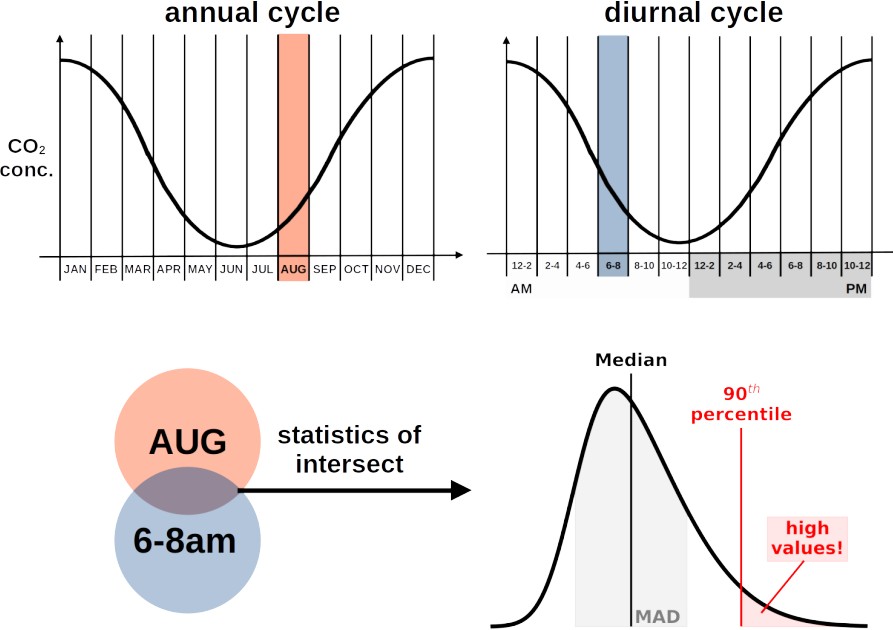

**Figure 2.** Illustration of our statistical analysis. Reference median, MAD (median absolute deviation), and $90^{th}$ percentile values are calculated for subsets of data points grouped by month and time of day (2-hourly). Individual measurements are compared to the statistics of the corresponding month-hour group. This calculation yields 12-x-12 reference statistics for each year. Details see Sec. 2.3.1.

### 2.3.2 Separation by atmospheric stability (transport distance)

We separate our dataset by atmospheric stability in an attempt to distinguish between long- and short-distant transport and evaluate if high concentration/flux observations (see Sec. 2.3.1) are more likely caused by distant or local sources. For this
separation we use the prevailing ABL heights (Sec. 2.2.3) and local vertical potential temperature gradients ($\frac{\partial \Theta}{\partial z}$ thereafter).
Assuming the ABL heights derived from ALC observations at Payerne provide a suitable characterisation of the regional-scale ABL dynamics, we combine these ABL heights with the local $\frac{\partial \Theta}{\partial z}$ profile to ensure atmospheric stability effects on shallow layers (below EC sensor height) are represented accurately. The $\frac{\partial \Theta}{\partial z}$ samples the very lowest surface layer (tower base to 212 m above ground) and can be used to identify strong surface temperature inversions.
Assuming the ABL to act as a lid and governing the mixing volume around the tower, we argue that an ABL below the sensor height (tower top) "cuts off" emission sources in the vicinity. This means that if the ABL lies between the tower top and the tower base, the air mass reaching the sensor at the top of the tower should be detached from emission sources in the vicinity and observations should be mainly governed by distant sources. In case the ABL is even below the tower base (tower is located on a hill top; see Sec. 2.1), also sources from the immediate vicinity around the tower could contribute. With this
in mind, we classify observations during an ABL below the tower top and unstable atmospheric conditions as well mixed and mainly influenced by *DISTANT* sources (Fig. 3, lower left quadrant). On the other hand, data points during a low ABL and





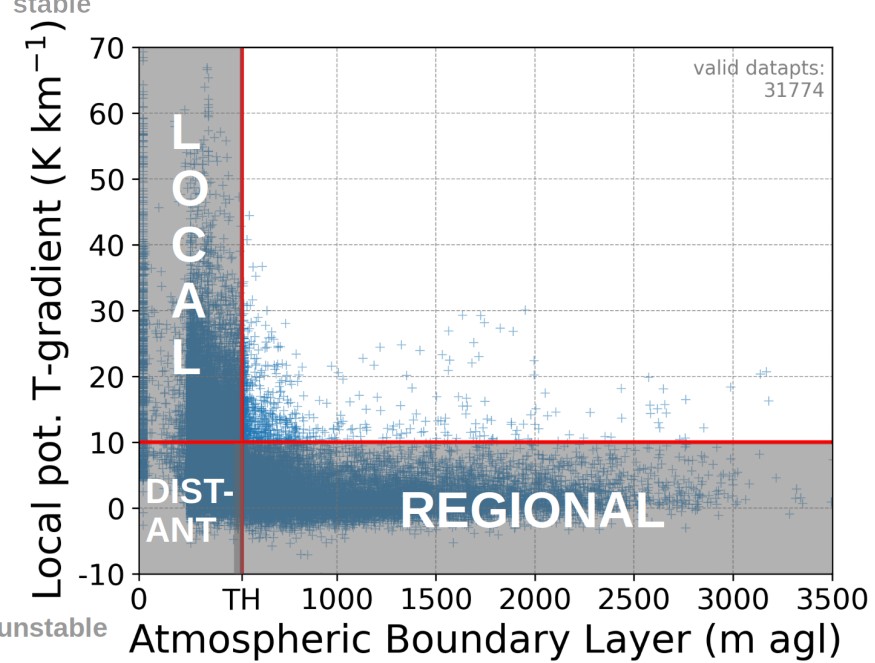

**Figure 3.** Scatter plot of all data points in the Atmospheric Boundary Layer (ABL) height vs. local potential temperature gradient ($\frac{\partial \Theta}{\partial z}$; at the tower) domain. The four quadrants correspond to: (upper left) ABL height below the tower height (TH) and a strong local temperature inversion (T-inv) — thereafter called *LOCAL* subset; (upper right) ABL height > TH and a strong T-inv; (lower left) ABL height < TH and a moderate/no T-inv — *DISTANT* subset; (lower right) ABL height > TH and a moderate/no T-inv — *REGIONAL* subset. The provided ABL height is relative to Payerne ground level (490 m asl; where the ABL is measured), not the Beromünster tower base (798 m asl). TH is therefore 520 m above ground level Payerne (212 m + 308 m altitude difference).

stable atmospheric conditions (Fig. 3, upper left quadrant) are assumed to be strongly affected by *LOCAL* sources because the strong surface inversion hinders vertical mixing and transport to the sensor, minimizing the influence of pollution signals from greater distances. Furthermore, the atmospheric conditions in the *REGIONAL* source selection (Fig. 3, lower right quadrant) allow for vertical mixing and provide a sufficient vertical extent of the ABL for horizontal transport.


### 2.3.3 Separation by wind direction, season, and time of day

Additionally to the separation by atmospheric stability (associated with transport distance), we separate the data by wind direction to provide another distinction between potential source regions (Fig. 4). Finally, we separate also by season and time of day, in order to get indications on how differently emission sources behave on a diurnal-seasonal basis (Fig. 4; e.g., 1 pm refers to 1:00 pm to 1:59 pm). For the separation by time of day we choose time slots of equal length (4 hours) and try to avoid






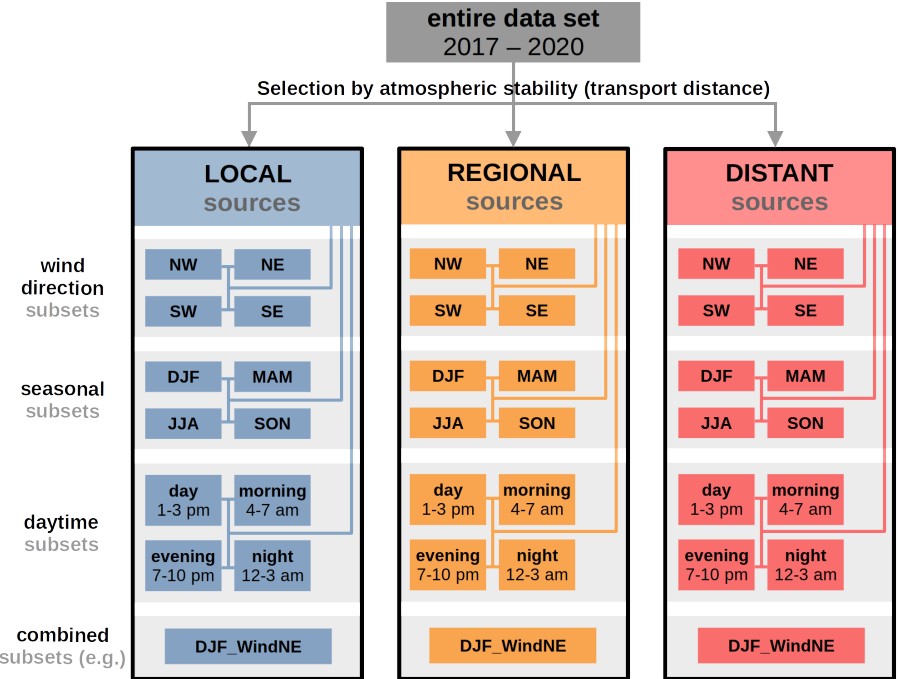

**Figure 4.** Flowchart of the subsets analyzed. Initially we split the dataset based on atmospheric stability to get a separation by transport distance (based on ABL height and $\frac{\partial \Theta}{\partial z}$) — *LOCAL*, *REGIONAL*, *DISTANT* source regions (Sec. 2.3.2). We then analyze source subsets separated by — wind direction, seasonality, time of day (Sec. 2.3.3).

the day-night transition for *day* and *night* as good as possible. As a result we neglect some hours for the time of day subsets (used hours listed in Fig. 4).

Since our statistical approach should eliminate diurnal and seasonal signals (e.g., also incl. weekday/weekend variations of anthropogenic $CO_2$ emissions), deviations from 10 % of high observations in different atmospheric stability and wind
direction subsets need to be related to changes in the sampling of source regions (Sec. 2.3.1). Performing the separation in *DJF/JJA/MAM/SON* and *day/night/morning/evening* for the entire dataset, i.e., no separation by source region, show high observations close to 10 % (Fig. A1), indicating a sufficient removal of the diurnal and seasonal signal.

## 2.4   Flux Footprints (FFPs)

The footprint shows the field of view of a flux/concentration sensor, i.e., the area sampled by the sensor. It is an attempt to
visualize the contribution of surface source/sink regions (mostly upwind) on measured turbulent fluxes/concentrations (Vesala et al., 2008) and its size is depending on measurement height, wind direction, surface roughness, and atmospheric stability (Leclerc and Thurtell, 1990).

In a Lagrangian perspective, virtual particles can be traced backward in time from the sensor to any potential surface source/sink (intersect with the surface). The particle touchdown locations and velocities can be sampled and further be used to





derive mean fluxes at the measurement location and further FFPs (Vesala et al., 2008). This step can for example be done with a Lagrangian particle dispersion model (LPDM) or an LPDM-derived parameterization (e.g., Kljun et al., 2015).

Concentration footprints (CFPs) can also be derived with a LPDM model while considering back-traced particles not just at their surface touchdown, but sampling the time the virtual particles interact with the surface layer along their back-trajectories (e.g., are residing within the lowest 100 m atmospheric layer; Oney et al., 2015). Particles are assumed to be influenced by surface sources/sinks while staying in this lowest atmospheric layer.

To conclude, FFPs on the one hand show the (rather local) sources/sinks regions upwind of the sensor to which turbulent EC-fluxes can be traced back directly and contribute to observed fluxes. These fluxes ($\mu$mol m$^{-2}$ s$^{-1}$) influence the concentrations ($\mu$mol mol$^{-1}$) at the sensor on a short time scale of sub-second to minutes. CFPs on the other hand show (larger scale) sources/sinks regions influencing the concentration advected air parcels (hourly, daily, seasonal scale signals). Direct turbulent fluxes can not be traced on such long distances. CFPs can be seen as the region where the (averaged) background signal originates from, while FFPs show the region influencing the observations on a shorter temporal scale.

For this paper we perform a FFP estimate with a two-dimensional parameterization (Kljun et al., 2015) for the different data subsets (Sec. 2.3), e.g., winter (*DJF*) and summer (*JJA*). The output of this parameterization is a spatial map of the area sampled by the tower, expressed as isolines showing the contribution percentages to the measured fluxes (Fig. 8). The parameterization is derived from an ensemble of backward LPDM simulations and is valid for a broad range of boundary layer conditions and measurement heights (Kljun et al., 2015).

Most input variables required to calculate the source area are taken from the EddyPro Software output (Sec. 2.2.2; mean wind speed, Obukhov length, standard deviation of the across-wind velocity, friction velocity) in addition to the wind direction, the measurement height, and measurements/estimates of the ABL height (Sec. 2.2.3). The FFP estimate can only be derived for ABL heights above the flux sensor height, since surface fluxes become decoupled from the surface once the ABL height is below the sensor (ABL acting as a lide; Kljun et al., 2015). Therefore, observations with a ABL height below the sensor are excluded from the shown FFP climatology estimate. However, data points measured during a lower-than-sensor ABL are still used for our analysis of high concentration/flux observations and the correlation analysis discussed in the result section.

## 3 Results and discussion

### 3.1 High concentration/flux observations

One of our goals is to identify the origin of high trace gas and flux observations at Beromünster tower. Therefore, we separate the observations mainly twofold, according to (1) atmospheric stability (enhancing/damping atmospheric transport/mixing; associated with source distance from the tower) and (2) wind direction (see Methods Sec. 2.3.2 and 2.3.3), in order to investigate observation subsets originating from different source regions. Additionally, we also separate by season and time of day, to get indications on diurnal-seasonal variations of different emission sources. Values greater than $\sim 10$ % in Fig. 5 and 6 indicate an above average number of high observations in the respective subsets (compared to the statistics of the entire dataset; Fig. A1; see definitions in Sec. 2.3.1).





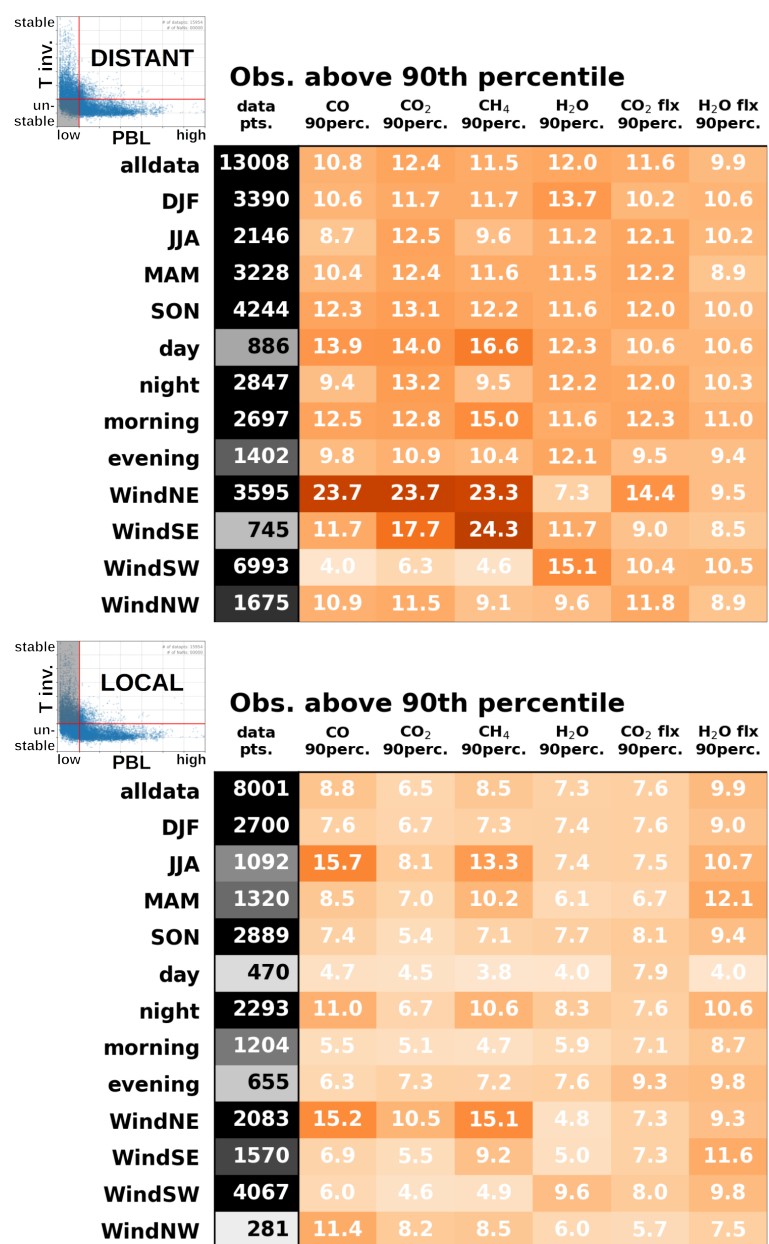

**Figure 5.** Percentages of high CO, $CO_2$, $CH_4$, $H_2O$ concentrations and $CO_2$, $H_2O$ fluxes in various subsets in the *DISTANT* (top) and *LOCAL* (bottom) source selections (Sec. 2.3.2). Percentages greater than ∼ 10 % indicate an above average number of high observations in the respective subsets (Sec. 2.3.1). Similar and more extensive figures for all four quadrants shown in Fig. 3 are provided in the appendix (Fig. A1, A2, A3, A4, and A5).





**Transport distance**

In general, the *DISTANT* source subset is more likely to see high CO, $CO_2$, $CH_4$, and $H_2O$ concentrations and $CO_2$ fluxes

($> 10\%$ in most subsets in Fig. 5 top). The *LOCAL* source subset on the other hand shows predominately fewer high concentration and $CO_2$ flux observations ($< 10\%$ of observations in Fig. 5 bottom). For $H_2O$ fluxes there are no distinct differences between *DISTANT* and *LOCAL* source subsets. Fewer high observations in the *LOCAL* source subset indicate that pollution events at Beromünster are mostly not caused by local sources, but the pollution is transported from further away to the tower (also elevated percentages of high observations in the *REGIONAL* source subset; Fig. A4).

While the seasonal signal is removed with our statistical method when averaging over the entire dataset, we still see a difference between winter (*DJF*) and summer (*JJA*) comparing *DISTANT* (Fig. 5; top) and *LOCAL* (Fig. 5; bottom) sources. Especially, $CH_4$ and CO concentrations are elevated in the *LOCAL JJA* subset by $\sim 30$ and $60\%$ ($\sim 13$ to $16\%$ compared to the expected $10\%$), respectively, while in the rest of the year high observations are below average ($< 10\%$). While most species show slightly elevated numbers of high observations for all seasons in the *DISTANT* source selection.

Similarly, the diurnal cycle is also removed with our statistical method when averaging over the entire dataset. However, again the diurnal cycle appears to be different between *DISTANT* (Fig. 5; top) and *LOCAL* (Fig. 5; bottom) sources. Particularly, the differences for *morning* and *day* in the two source subsets are large; $\sim 13\text{-}17\%$ of high CO, $CO_2$, and $CH_4$ concentrations in the *DISTANT* subset vs. only $\sim 4\text{-}5\%$ in the *LOCAL* subset. The differences during *evening* and *night* are less pronounced.

**Wind direction**

However, the strongest variability in high observations we see depending on the prevailing wind direction (*WindNE*, *WindSE*, *WindSW*, *WindNW* in Fig. 5 top and bottom). CO, $CO_2$, and $CH_4$ concentrations are more than twice as likely to be categorized as high during wind from the NE for *DISTANT* sources ($\sim 23\%$ vs. expected $10\%$ if averaged over all wind directions; *alldata* in Fig. A1). For *LOCAL* sources this pattern is less pronounced, $\sim 15\%$ for CO and $CH_4$, $\sim 11\%$ for $CO_2$ for *WindNE*. For

the *DISTANT WindNE* $CO_2$ fluxes are also elevated (14.4 %), while below average in the *LOCAL WindNE* (7.3 %). Also the *DISTANT WindSE* subset shows elevated percentages of high concentrations, especially $CH_4$ with a value of $\sim 24\%$.

As a side note, the seasonal wind direction distribution remains relatively constant over the year. The main wind axis is oriented in the SW-NE axis, with roughly $50\%$ of the cases with wind from the SW (Fig. 1a). The ratio of cases with SW wind varies between $\sim 30$ and $\sim 60\%$ in spring (*MAM*) and winter (*DJF*), respectively. And slightly more NE wind is

observed during spring (*MAM*); $\sim 30\%$ vs. 15-20 % in the other seasons (supplementary wind roses in Fig. S3). Concerning the diurnal variability, the percentage of cases with SW varies from 45-50 % during *morning/day/night* and $\sim 35\%$ in the *evening* (supplementary wind roses in Fig. S4).

Figure 6 combines the information on transport distance (*DISTANT*, *REGIONAL*, *LOCAL*) and wind direction (*NW*, *NE*, *SE*, *SW*). CO, $CO_2$, and $CH_4$ concentrations clearly show more high concentrations during easterly winds (*NE*, *SE*; values

$> 10\%$) and longer transport distances (*DISTANT*, *REGIONAL*; circle segments further away from the center). While this





signal is strongest for *NE* for CO, $CO_2$ (> 20% for *DISTANT NE*), $CH_4$ concentrations show the strongest enhancement of high concentrations for *DISTANT* and *REGIONAL SE* with $\sim$ 24-27%. For the $CO_2$ fluxes the easterly wind directions are less pronounced, however *DISTANT NE* show $\sim$ 14 % of high fluxes. $H_2O$ concentrations show a tendency for elevated concentrations for westerly winds, especially *DISTANT SW* with $\sim$ 15 % (Sempacher lake is located in this direction). The

pattern for the $H_2O$ fluxes is inconsistent and does not show any clear enhancements.

Figure 6 again emphasizes that only few of the strongest emission sources are located in the immediate vicinity of Beromün-ster tower (*LOCAL* mostly below 10 %; only elevated for CO and $CH_4$). And that the strongest CO, $CO_2$, and $CH_4$ emitters are likely located further away in the NE and SE of the tower. This fits well with the location of the Zurich metropolitan area in the NE (outskirts roughly 30 $km$ away; Fig 1c) and Luzern in the SE (roughly 15 $km$ away), respectively. The quadrant SW

of Beromünster tower on the other hand seems to be the origin of comparably clean air (consistently smaller than 10 % for CO, $CO_2$, and $CH_4$ concentrations). This is consistent with the surroundings, as this area is less urbanized than areas to the E of the tower. Furthermore, the SW quadrant includes a partly environmentally protected area ("UNESCO Biosphärenpark Entlebuch"; 10 - 15 km distance).

The weaker enhancement of high $CO_2$ fluxes from the NE can be associated to the long transport from most potential emitter

to the tower. Although our analysis indicates a FFP that might sometimes also include the outskirts of Zurich (see Sec. 3.3), the relative contribution to direct $CO_2$ fluxes from Zurich arriving at the tower will be small and therefore difficult to detect (the area between FFP isolines increases with distance (Fig. 8), i.e., the per-square-meter-contribution decreases), and therefore difficult to detect.

### 3.2   Correlations between trace gas concentrations

In order to investigate the relationship between different species emission sources, we perform an inter-species correlation analysis for CO, $CO_2$, and $CH_4$ concentrations and similar to previous studies (with shorter time series; Satar et al., 2016; Berhanu et al., 2017) we find a strong correlation between all three species at Beromünster during winter (*DJF*, Fig. 7b, $R^2 \sim 0.8$), while a very limited correlation in summer for CO vs. $CH_4$ (*JJA*, Fig. 7c, $R^2 \sim 0.4$) and no inter-species correlation for the other two combinations ($R^2 \sim 0$ for $CO_2$ vs. $CH_4$ and CO vs. $CO_2$) — also see Fig. A7, and A6. Additionally, we

look at the correlations for different wind sectors in winter (*DJF_WindSW*, *DJF_WindNE*) — Fig. 7d and e — and find weaker correlations for *DJF_WindSW* than for *DJF_WindNE* ($R^2$ of 0.68 - 0.80 and 0.84 - 0.93, respectively, for CO vs. $CH_4$). Since *DJF_WindSW* is less prone to high values (see Fig. 5) this weaker correlation might indicate that the process(es) causing the lower concentrations is (are) also less correlated than the process(es) causing higher concentrations in *DJF_WindNE*.

Since the main CO source is incomplete combustion (of fossil fuels; resulting in co-located $CO_2$ emissions), it is no surprise

that CO and $CO_2$ (Fig. A7) are strongly correlated during winter, when the biosphere is contributing less to $CO_2$ variability and CO and $CO_2$ emissions can be expect to peak due to additional combustion for heating purposes. In summer, the lack of correlation is probably related to the increased biosphere activity (photosynthesis dominates over respiration) and the stronger atmospheric mixing, making potential emission correlations harder to detect.





The strong correlation between $CO_2$ and $CH_4$ in winter (Fig. 7b) is not straight forward to explain due to no overlap of
emission sources — CO: incomplete combustion; $CH_4$: mainly pasture farming, pipeline leakage, outgassing of landfills, and
wetlands. Again, CO emissions can be expected to peak during the heating period. While anthropogenic $CH_4$ emissions are
relatively constant over the year (Hiller et al., 2014) with a tendency to smaller emissions in the cold season, especially in
the agricultural areas around the Beromünster site (Henne et al., 2016). Satar et al. (2016) also investigated inter-species
correlations at Beromünster and find a somewhat smaller correlation between CO and $CH_4$ for the winter months ($R^2 \sim 0.75$;
2012 - 2014; 212.5 m inlet, same as used here) and a height dependency of the correlation at Beromünster tower ($R^2 \sim 0.55$
for the 12.5 m inlet). They argue that since the atmospheric mixing should be independent of species, the correlation height
dependency may be a reflection of differences in the relative importance of local versus distant sources. However, by comparing
the corresponding CO-$CH_4$-correlations for *DJF_WindNE* in our *LOCAL* and *DISTANT* source subsets (not shown), we find
very similar values (*LOCAL*: $R^2 = 0.88$-$0.93$; *DISTANT*: $R^2 = 0.85$-$0.95$), only in the *REGIONAL DJF_WindNE* subset the
correlations are lower ($R^2 = 0.46$-$0.88$). We conclude that the processes leading to this strong winter CO-$CH_4$-correlation are
likely related to atmospheric mixing of high concentration air masses on their way to the tower. However further investigation
is needed to better understand temporal and spatial variability of the corresponding emission sources and transport pathways
to the tower.

Correlations between $CO_2$ and $CH_4$ in spring (*MAM*; not shown) and in fall (*SON*; not shown) are somewhere between
summer and winter. This indicates that the causing process might be a winter-process spreading out into the shoulder seasons,
or a all-season-process where the correlation is weakened in summer due to atmospheric mixing, or a combination of these two
types of processes.

### 3.3 Characterization of the Beromünster FFP

As an addition for the interpretation of the EC-flux observations, we present climatologies of parameterized flux footprints
(FFPs; Sec. 2.4 and Fig. 1b) of the highest Beromünster observation level (212 m agl). Averaged for the entire dataset the FFP
shape (*alldata*; Fig. 8a) is elongated in the main wind axis (NE-SW). In this along-wind direction the extent of the FFP is
$\sim 50$ km, while in the perpendicular across-wind direction it's $\sim 25$ km — the extent referring to the 80 % isoline enclosing
the area where 80 % of the observed EC-fluxes are estimated to originate from. The relatively large spatial extent of the FFP
is a result of the great observation height. However, we want to emphasize that the relative contribution of emission sources
to observed EC-fluxes decreases with the distance from the tower, i.e., the area between consecutive isolines increases with
distance from the tower, resulting in a smaller per-area-contribution to the observed fluxes.

The winter FFP (*DJF*; Fig. 8b) is even more elongated and slightly more narrow in the across-wind direction ($\sim 100$-x-
25 km), with an otherwise similar shape. This is mainly a result of higher winter wind speeds (not shown; Fig. S3). The
summer FFP (*JJA*; Fig. 8c) has a rounder shape with a smaller horizontal extent ($\sim 30$-x-20 km), which is a result of lower
365 summer wind speeds and a sightly more evenly distribution of wind direction in this season (not shown; Fig. S3). Finally, the
FFP for winter with prevailing wind from the NE (*DJF_WindNE*; Fig. 8d) extends roughly 30 km towards the NE which might
indicate a sampling of emission sources in the vicinity of Zurich.



Although, the presented Beromünster FFPs are exceptionally large (compared to traditional flux towers for micro-meteorological applications), the corresponding concentration footprints (CFPs; previously published; Oney et al., 2015, Fig. 12 therein) ex-

370   tend over all of Switzerland (winter; $\sim$ 400 km along-wind) and beyond (summer; $\sim$ 700 km along-wind), but limited by the Alps in the South.



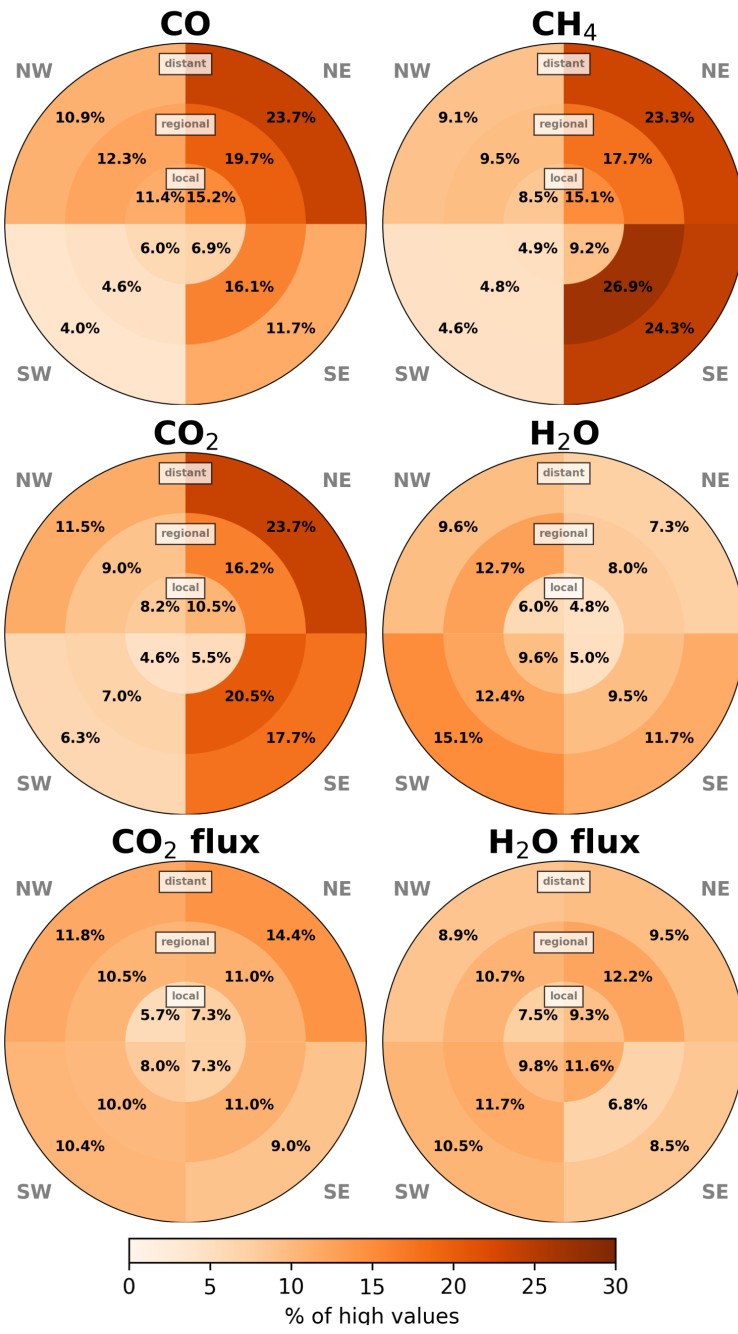

**Figure 6.** Percentages of high concentrations/fluxes for data subsets divided by wind direction (four quadrants: wind from NE, SE, SW, or NW) and atmospheric stability (associated with transport distance; Sec. 2.3.2; illustrated as the distance from the tower in the center) for CO, $CH_4$, $CO_2$, and $H_2O$ concentrations, and $CO_2$ and $H_2O$ fluxes. Percentages higher/lower than $\sim 10$ % indicate a greater/smaller number of high observations in the respective subsets (Sec. 2.3.1).



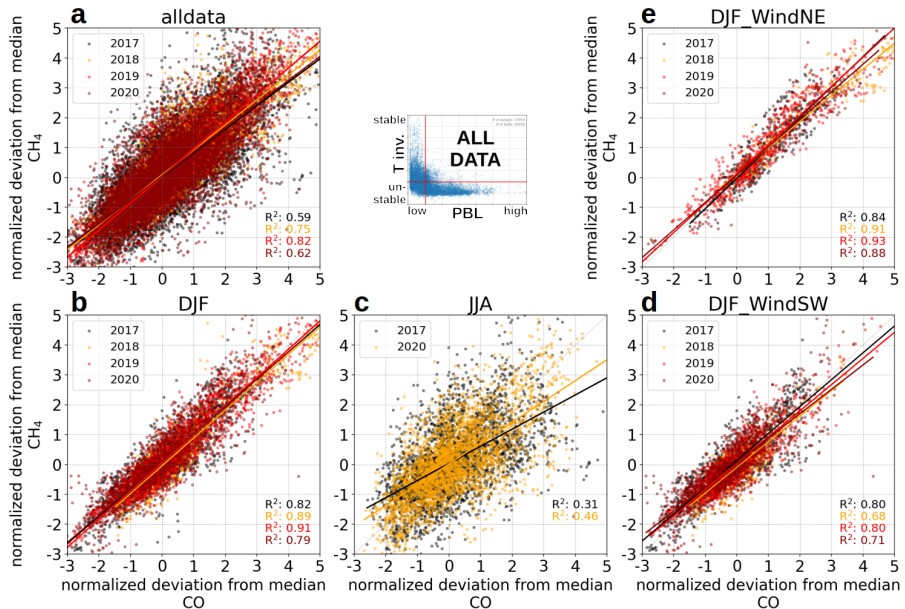

**Figure 7.** Correlation of CO versus $CH_4$ concentrations for (a) all data points (*alldata*), (b) winter (*DJF*), (c) summer (*JJA*), (d) *DJF_WindSW*, and (e) *DJF_WindNE* without any source (atmospheric stability) selection, i.e., based on the entire dataset. Different years are color-coded. The concentrations are normalized (by the median absolute deviation; MAD; see Sec. 2.3.1). Similar figures for $CO_2$ vs. $CH_4$ and $CO_2$ vs. CO are provided in the appendix (Fig. A6 and A7).

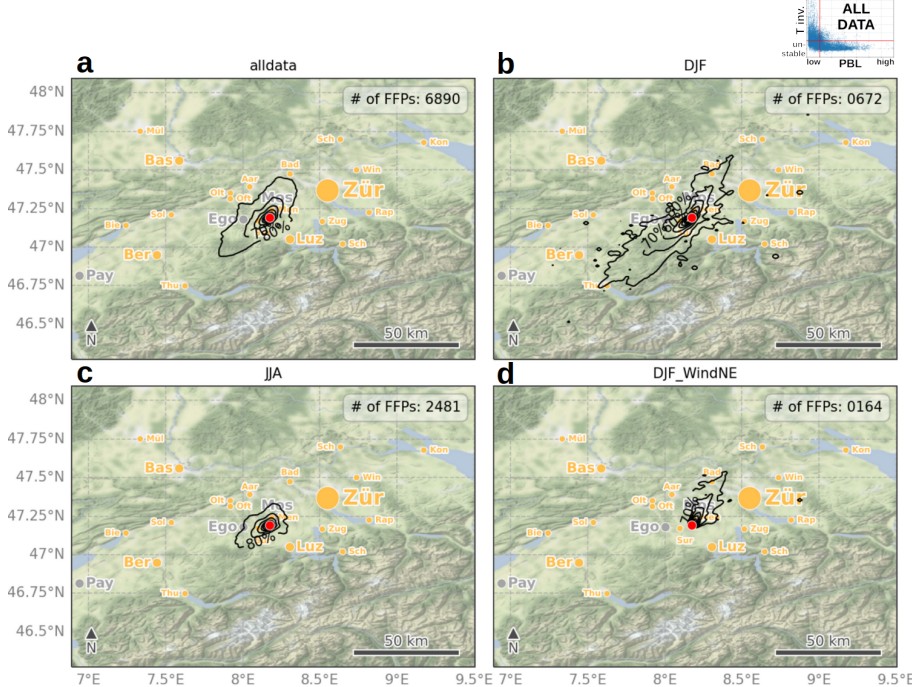

**Figure 8.** Parameterized flux footprint (FFP) climatologies of the Beromünster tall tower for the entire dataset. (a) all data points (*alldata*), (b) winter (*DJF*), (c) summer (*JJA*), and (d) winter with prevailing wind from the NE (*DJF_WindNE*). Isolines indicate the relative contribution to the observed fluxes, i.e., the 80 %-isoline encloses the area from which 80 % of the observed direct fluxes are estimated to originate from: Beromünster tower (red dot); Other measurement stations (gray): Egolzwil (Ego), Mosen (Mos), Payerne (Pay); Urban areas (yellow): Aarau (Aar), Baden (Bad), Basel (Bas), Ber (Bern), Biel (Bie), Konstanz (Kon), Luzern (Luz), Menziken (Men), Mülhausen (Mül), Oftringen (Oft), Olten (Olt), Rapperswil-Jona (Rap), Schaffhausen (Sch; NE), Schwyz (Sch, SE), Soloturn (Sol), Sursee (Sur), Thun (Thu), Winterthur (Win), Zug (Zug), Zürich (Zür). Map tiles by Stamen Design under CC BY 3.0. Data by © OpenStreetMap under ODbL.



## 4 Conclusions and Outlook

We present an analysis of four years (2017 - 2020) of CO, $CO_2$, $CH_4$, $H_2O$ concentrations and $CO_2$, $H_2O$ fluxes from the Swiss Beromünster tall tower focusing on transport distance and wind direction of potential emission sources, inter-species correlations, and a characterization of the so-called flux footprints (FFPs), indicating the origin of direct Eddy Covariance (EC)-fluxes. We find that:

- pollution events (CO, $CO_2$, $CH_4$ concentrations) at Beromünster tower are less associated with emission sources in its immediate vicinity, but are likely transported from further away; predominately from the NE (Zurich metropolitan area) and SE (Luzern).

- previously published Concentration Footprints (CFPs; Oney et al., 2015) indicate a sensitivity of the observed concentrations to emissions from 100s of kilometers around Beromünster tower (all over Switzerland and beyond; limited by the Alps in the South).

- while the parameterized Flux Footprints (FFP) of Beromünster presented here, indicate a direct influence of emission fluxes in the surrounding 20-50 km of the tower (depending on season; smaller more round summer FFP, larger more elongated (NE-SW) winter FFP). Fluxes originating from close urban areas might reach the tower at least occasionally.

- the strongest enhancement of high $CH_4$ concentrations is found during SE wind (high concentrations $\sim 2.5$ times more likely than compared to entire dataset).

- the corresponding signal of high $CO_2$ fluxes is weaker (less enhanced than for $CO_2$ concentrations) and limited NE wind; probably a result of the long distance from the strongest $CO_2$ flux sources (likely located near/in Zurich).

- a strong correlation between CO, $CO_2$, and $CH_4$ concentrations in winter ($R^2 \sim 0.8$), while a very limited summer correlation for CO vs. $CH_4$ ($R^2 \sim 0.4$) and no summer correlation for the other two species combinations ($R^2 \sim 0$ for $CO_2$ vs. $CH_4$ and CO vs. $CO_2$).

- CO, $CO_2$, and $CH_4$ concentrations are weaker correlated in winter during SW wind ($R^2 \sim 0.7$) than during NE wind ($R^2 \sim 0.9$).

**Outlook**

While we use the ABL data derived from observations at Payerne in this analysis under the assumption of regionally representative boundary layer dynamics, observations from an ALC operated at Nottwil (47.142°N, 8.131°E 510 m asl), a site in just 6 km distance from the Beromünster tower could be exploited in future analysis. At Nottwil, MeteoSwiss is gathering attenuated backscatter profiles using a Vaisala CL31 since March 2022. This ALC model has better capabilities for the detection of shallow layers. Using the automatic CABAM algorithm (Kotthaus and Grimmond, 2018), layer heights as low as 50 m can be detected. Given the lower signal-to-noise of this ALC, it has greater uncertainty for detection of layer heights $> 1$km compared

to the ALC used in the current study, but this would have a limited implication for the source area assessment approach applied here.

An additional analysis could be done based on a separation by sunshine duration, since biospheric activity is highly corre-
lated with available sunlight. Furthermore, an extension to the present study is planned to investigate the relationship between FFPs and CFPs to give an update and further analysis to the previously published CFPs (Oney et al., 2015; Rust et al., 2022) by investigating various meteorological input datasets (e.g., different spatial resolution) in combination with a multitude of atmospheric transport model settings (Lagrangian atmospheric model FLEXPART; Pisso et al., 2019; Stohl et al., 2005). We aim to use these atmospheric transport simulations in combination with earlier published Swiss $CO_2$ and $CH_4$ emission inven-
tories (Albrecht et al., 2012; Hiller et al., 2014) to simulate the Beromünster concentration and flux observations. Furthermore, we plan to derive FFPs with the FLEXPART model rather than the parameterization presented here, which will allow for a consistent comparison of the two footprint types.

*Code and data availability.* EddyPro Software v7.0.6 is a commercial software available from LI-COR Biosciences (https://www.licor.com/env/support/EddyPro/software.html); REddyProc v1.2 (Wutzler et al., 2018) is freely available as an online tool here: https://www.bgc-jena.mpg.de/REddyProc/brew/REddyProc.rhtml; A collection of the data used for the analysis in this study is available here: https://doi.org/10.25365/phaidra.361.

## Appendix A: Additional figures

*Author contributions.* AP and ML designed the study. AP performed the analysis and wrote the manuscript with contributions from all authors. SK and RR processed the ALC data to retrieve the ABL heights.

*Acknowledgements.* We thank MeteoSwiss for providing global radiation data from Egolzwil and Mosen used in this study and Rüdiger Schander, Peter Nyfeler, and Michael Schibig for their technical assistance at Beromünster. AP was supported by computational and storage resources through the ECMWF special project *spatvojt*. AP and ML were supported by the Swiss National Science Foundation project SNF 172550. We further thank the members of the Flexteam (flexteam.univie.ac.at) at the University of Vienna for the valuable discussions, their support, and the great working environment - Katharina Baier, Lucie Bakels (proof reading), Silvia Bucci (technical help), Marina Dütsch (proof reading), Ioanna Evangelou, Petra Seibert, Andreas Stohl, Rakesh Subramanian, Daria Tatsii, and Martin Vojta.





**Obs. above/below 90th/10th percentile**

ALL DATA

| | data pts. | CO 90perc. | CO₂ 90perc. | CH₄ 90perc. | H₂O 90perc. | CO₂ flx 90perc. | H₂O flx 90perc. | CO 10perc. | CO₂ 10perc. | CH₄ 10perc. | H₂O 10perc. | CO₂ flx 10perc. | H₂O flx 10perc. |
|---|---|---|---|---|---|---|---|---|---|---|---|---|---|
| alldata | 31774 | 10.4 | 10.4 | 10.4 | 10.4 | 10.1 | 10.4 | 10.4 | 10.4 | 10.4 | 10.4 | 10.3 | 10.3 |
| DJF | 7632 | 10.5 | 10.5 | 10.5 | 10.5 | 9.6 | 10.5 | 10.5 | 10.5 | 10.5 | 10.5 | 10.4 | 10.2 |
| JJA | 6397 | 10.4 | 10.4 | 10.4 | 10.4 | 10.4 | 10.3 | 10.4 | 10.4 | 10.4 | 10.4 | 10.4 | 10.3 |
| MAM | 7901 | 10.5 | 10.5 | 10.5 | 10.5 | 10.5 | 10.4 | 10.5 | 10.5 | 10.5 | 10.5 | 10.4 | 10.5 |
| SON | 9844 | 10.4 | 10.4 | 10.4 | 10.4 | 10.1 | 10.3 | 10.4 | 10.4 | 10.4 | 10.4 | 10.2 | 10.3 |
| day | 5747 | 10.4 | 10.4 | 10.4 | 10.4 | 10.2 | 10.4 | 10.4 | 10.4 | 10.4 | 10.4 | 10.4 | 10.4 |
| night | 5382 | 10.4 | 10.4 | 10.4 | 10.4 | 10.1 | 10.4 | 10.4 | 10.4 | 10.4 | 10.4 | 10.3 | 10.2 |
| morning | 5635 | 10.5 | 11.0 | 11.0 | 10.4 | 10.7 | 10.9 | 10.3 | 9.7 | 10.3 | 10.5 | 10.2 | 10.5 |
| evening | 4829 | 10.4 | 10.4 | 10.4 | 10.4 | 10.1 | 10.4 | 10.4 | 10.4 | 10.4 | 10.4 | 10.4 | 10.4 |
| WindNE | 9096 | 20.2 | 17.9 | 19.3 | 7.1 | 11.4 | 10.3 | 3.3 | 7.2 | 4.1 | 13.6 | 10.9 | 8.1 |
| WindSE | 2942 | 9.4 | 10.8 | 15.4 | 7.3 | 8.3 | 9.9 | 11.5 | 10.8 | 14.9 | 16.1 | 9.3 | 13.7 |
| WindSW | 15827 | 4.7 | 6.1 | 4.7 | 12.8 | 9.6 | 10.7 | 15.1 | 11.8 | 13.9 | 8.0 | 9.9 | 11.0 |
| WindNW | 3909 | 11.5 | 10.2 | 9.2 | 10.9 | 10.7 | 9.7 | 7.1 | 12.2 | 7.8 | 8.6 | 11.7 | 10.1 |
| DJF_WindNE | 1599 | 29.1 | 31.5 | 27.3 | 3.8 | 13.0 | 10.8 | 2.4 | 3.2 | 3.9 | 13.3 | 11.1 | 6.9 |
| DJF_WindSE | 636 | 11.3 | 11.9 | 16.7 | 7.1 | 7.9 | 6.1 | 10.2 | 12.9 | 12.9 | 22.6 | 10.2 | 13.2 |
| DJF_WindSW | 5001 | 4.1 | 3.3 | 4.2 | 13.7 | 8.9 | 11.0 | 13.8 | 12.9 | 12.8 | 8.5 | 10.1 | 10.9 |
| DJF_WindNW | 396 | 14.4 | 13.6 | 12.6 | 2.5 | 7.8 | 9.3 | 1.8 | 5.6 | 4.5 | 5.3 | 11.9 | 10.1 |
| JJA_WindNE | 1481 | 14.7 | 7.1 | 20.2 | 6.7 | 8.4 | 10.9 | 3.3 | 15.1 | 3.6 | 10.9 | 10.1 | 7.4 |
| JJA_WindSE | 585 | 10.1 | 11.1 | 16.4 | 7.9 | 9.6 | 11.1 | 8.9 | 9.6 | 18.6 | 8.5 | 10.4 | 15.0 |
| JJA_WindSW | 3126 | 8.5 | 13.1 | 5.6 | 11.1 | 11.2 | 10.5 | 14.7 | 7.3 | 13.5 | 9.8 | 10.0 | 10.5 |
| JJA_WindNW | 1205 | 10.0 | 7.1 | 7.7 | 14.2 | 11.0 | 8.7 | 8.5 | 13.1 | 6.6 | 12.3 | 11.8 | 11.4 |

**Figure A1.** same as Fig. 5 but for the entire dataset (no source selection; all four quadrants in Fig. 3; Sec. 2.3.2). Additionally showing intersects of DJF, JJA and wind subsets.

**Obs. above/below 90th/10th percentile**

DISTANT

| | data pts. | CO 90perc. | CO₂ 90perc. | CH₄ 90perc. | H₂O 90perc. | CO₂ flx 90perc. | H₂O flx 90perc. | CO 10perc. | CO₂ 10perc. | CH₄ 10perc. | H₂O 10perc. | CO₂ flx 10perc. | H₂O flx 10perc. |
|---|---|---|---|---|---|---|---|---|---|---|---|---|---|
| alldata | 13008 | 10.8 | 12.4 | 11.5 | 12.0 | 11.6 | 9.9 | 9.8 | 8.8 | 8.0 | 7.5 | 10.3 | 10.1 |
| DJF | 3390 | 10.6 | 11.7 | 11.7 | 13.7 | 10.2 | 10.6 | 9.1 | 8.9 | 6.9 | 4.5 | 11.2 | 10.3 |
| JJA | 2146 | 8.7 | 12.5 | 9.6 | 11.2 | 12.1 | 10.2 | 11.7 | 10.0 | 8.4 | 11.0 | 10.3 | 9.4 |
| MAM | 3228 | 10.4 | 12.4 | 11.6 | 11.5 | 12.2 | 8.9 | 10.4 | 8.5 | 9.6 | 8.2 | 10.4 | 10.8 |
| SON | 4244 | 12.3 | 13.1 | 12.2 | 11.6 | 12.0 | 10.0 | 8.9 | 8.4 | 7.5 | 7.6 | 9.7 | 9.8 |
| day | 886 | 13.9 | 14.0 | 16.6 | 12.3 | 10.6 | 10.6 | 7.0 | 5.6 | 5.8 | 5.5 | 9.7 | 11.9 |
| night | 2847 | 9.4 | 13.2 | 9.5 | 12.2 | 12.0 | 10.3 | 11.6 | 10.4 | 9.5 | 8.3 | 11.2 | 9.8 |
| morning | 2697 | 12.5 | 12.8 | 15.0 | 11.6 | 12.3 | 11.0 | 8.0 | 7.6 | 6.8 | 7.3 | 10.5 | 10.6 |
| evening | 1402 | 9.8 | 10.9 | 10.4 | 12.1 | 9.5 | 9.4 | 8.4 | 7.2 | 8.8 | 6.8 | 9.5 | 10.1 |
| WindNE | 3595 | 23.7 | 23.7 | 23.3 | 7.3 | 14.4 | 9.5 | 2.6 | 5.6 | 3.1 | 11.7 | 11.0 | 7.7 |
| WindSE | 745 | 11.7 | 17.7 | 24.3 | 11.7 | 9.0 | 8.5 | 8.5 | 11.7 | 8.3 | 9.1 | 8.6 | 10.6 |
| WindSW | 6993 | 4.0 | 6.3 | 4.6 | 15.1 | 10.4 | 10.5 | 14.4 | 9.4 | 10.8 | 4.9 | 10.4 | 11.3 |
| WindNW | 1675 | 10.9 | 11.5 | 9.1 | 9.6 | 11.8 | 8.9 | 6.5 | 12.2 | 6.9 | 8.6 | 9.7 | 10.0 |

**Figure A2.** same as Fig. 5 but for the *DISTANT* source selection (lower left quadrant in Fig. 3; Sec. 2.3.2). Intersecting subsets of season and wind not shown due to small sample size.

# References

Albrecht, S., Künzle, T., and Schaffner, B.: CO₂-Emissionskataster Schweiz. Aufbereitung von Datengrundlagen, Berechnung eines räumlich hoch aufgelösten Katasters, Tech. Rep. project number 11_096, Genossenschaft METEOTEST. Wetterprognosen, Erneuerbare Energien, Luft und Klima, Umweltinformatik, Fabrikstrasse 14, CH-3012 Bern, www.meteotest.ch, 2012.




| | data pts. | CO 90perc. | CO₂ 90perc. | CH₄ 90perc. | H₂O 90perc. | CO₂ flx 90perc. | H₂O flx 90perc. | CO 10perc. | CO₂ 10perc. | CH₄ 10perc. | H₂O 10perc. | CO₂ flx 10perc. | H₂O flx 10perc. |
|---|---|---|---|---|---|---|---|---|---|---|---|---|---|
| alldata | 8001 | 8.8 | 6.5 | 8.5 | 7.3 | 7.6 | 9.9 | 12.2 | 12.3 | 15.7 | 15.8 | 9.8 | 12.0 |
| DJF | 2700 | 7.6 | 6.7 | 7.3 | 7.4 | 7.6 | 9.0 | 13.7 | 13.7 | 16.6 | 19.8 | 9.4 | 11.1 |
| JJA | 1092 | 15.7 | 8.1 | 13.3 | 7.4 | 7.5 | 10.7 | 6.2 | 9.3 | 15.4 | 9.0 | 10.3 | 12.8 |
| MAM | 1320 | 8.5 | 7.0 | 10.2 | 6.1 | 6.7 | 12.1 | 10.4 | 12.8 | 13.1 | 15.5 | 9.5 | 11.2 |
| SON | 2889 | 7.4 | 5.4 | 7.1 | 7.7 | 8.1 | 9.4 | 14.0 | 11.9 | 16.2 | 14.8 | 10.1 | 12.8 |
| day | 470 | 4.7 | 4.5 | 3.8 | 4.0 | 7.9 | 4.0 | 22.8 | 16.0 | 25.3 | 26.8 | 8.5 | 14.3 |
| night | 2293 | 11.0 | 6.7 | 10.6 | 8.3 | 7.6 | 10.6 | 9.4 | 10.5 | 12.0 | 13.6 | 9.2 | 11.2 |
| morning | 1204 | 5.5 | 5.1 | 4.7 | 5.9 | 7.1 | 8.7 | 15.4 | 14.1 | 20.8 | 17.4 | 9.2 | 12.1 |
| evening | 655 | 6.3 | 7.3 | 7.2 | 7.6 | 9.3 | 9.8 | 16.6 | 15.4 | 18.0 | 19.7 | 13.1 | 13.7 |
| WindNE | 2083 | 15.2 | 10.5 | 15.1 | 4.8 | 7.3 | 9.3 | 6.1 | 9.4 | 8.0 | 19.7 | 10.9 | 9.7 |
| WindSE | 1570 | 6.9 | 5.5 | 9.2 | 5.0 | 7.3 | 11.6 | 13.9 | 11.1 | 19.4 | 21.0 | 9.5 | 14.2 |
| WindSW | 4067 | 6.0 | 4.6 | 4.9 | 9.6 | 8.0 | 9.8 | 14.9 | 14.3 | 18.5 | 12.4 | 9.4 | 12.1 |
| WindNW | 281 | 11.4 | 8.2 | 8.5 | 6.0 | 5.7 | 7.5 | 9.6 | 11.4 | 11.7 | 7.1 | 8.9 | 14.9 |

**Figure A3.** same as Fig. 5 but for the *LOCAL* source selection (upper left quadrant in Fig. 3; Sec. 2.3.2). Intersecting subsets of season and wind not shown due to small sample size.

| | data pts. | CO 90perc. | CO₂ 90perc. | CH₄ 90perc. | H₂O 90perc. | CO₂ flx 90perc. | H₂O flx 90perc. | CO 10perc. | CO₂ 10perc. | CH₄ 10perc. | H₂O 10perc. | CO₂ flx 10perc. | H₂O flx 10perc. |
|---|---|---|---|---|---|---|---|---|---|---|---|---|---|
| alldata | 9922 | 11.5 | 11.0 | 10.8 | 10.9 | 10.5 | 11.5 | 9.4 | 11.1 | 8.8 | 9.9 | 10.9 | 8.8 |
| DJF | 1293 | 17.2 | 16.6 | 15.3 | 9.1 | 12.5 | 13.2 | 7.2 | 8.1 | 6.8 | 7.1 | 10.3 | 7.2 |
| JJA | 2987 | 10.0 | 9.4 | 10.2 | 10.9 | 10.1 | 10.6 | 10.3 | 11.4 | 8.6 | 10.5 | 10.7 | 9.7 |
| MAM | 3224 | 11.4 | 10.0 | 9.6 | 11.5 | 10.4 | 11.2 | 10.5 | 11.5 | 10.3 | 10.4 | 11.0 | 9.5 |
| SON | 2418 | 10.3 | 11.2 | 10.6 | 11.0 | 9.8 | 12.0 | 8.1 | 11.8 | 8.2 | 10.1 | 11.3 | 7.8 |
| day | 4272 | 10.5 | 10.3 | 10.0 | 11.0 | 10.3 | 11.3 | 9.5 | 10.8 | 9.4 | 9.6 | 10.8 | 9.4 |
| night | 111 | 22.5 | 25.2 | 32.4 | 9.9 | 18.0 | 10.8 | 6.3 | 7.2 | 6.3 | 2.7 | 10.8 | 2.7 |
| morning | 1650 | 11.1 | 12.4 | 9.5 | 12.1 | 10.8 | 12.4 | 10.5 | 10.3 | 7.6 | 10.7 | 10.8 | 8.9 |
| evening | 2539 | 12.3 | 10.3 | 11.7 | 10.1 | 10.8 | 11.1 | 8.9 | 10.9 | 8.3 | 10.1 | 10.3 | 9.0 |
| WindNE | 3221 | 19.7 | 16.2 | 17.7 | 8.0 | 11.0 | 12.2 | 1.8 | 7.7 | 2.3 | 12.0 | 10.9 | 7.3 |
| WindSE | 409 | 16.1 | 20.5 | 26.9 | 9.5 | 11.0 | 6.8 | 4.9 | 8.1 | 6.4 | 10.5 | 10.5 | 13.4 |
| WindSW | 4377 | 4.6 | 7.0 | 4.8 | 12.4 | 10.0 | 11.7 | 16.4 | 13.3 | 14.2 | 8.7 | 9.5 | 9.3 |
| WindNW | 1915 | 12.3 | 9.0 | 9.5 | 12.7 | 10.5 | 10.7 | 7.2 | 12.5 | 7.9 | 9.1 | 14.1 | 9.3 |

**Figure A4.** same as Fig. 5 but for the *REGIONAL* source selection (lower left quadrant in Fig. 3; Sec. 2.3.2). Intersecting subsets of season and wind not shown due to small sample size.

Aubinet, M., Vesala, T., and Papale, D.: Eddy Covariance. A Practical Guide to Measurement and Data Analysis, Springer Atmospheric Sciences, Springer, Dordrecht, https://doi.org/10.1007/978-94-007-2351-1, 2012.

Bamberger, I., Oney, B., Brunner, D., Henne, S., Leuenberger, M., Buchmann, N., and Eugster, W.: Observations of Atmospheric Methane and Carbon Dioxide Mixing Ratios: Tall-Tower or Mountain-Top Stations?, Boundary-Layer Meteorology, 164, 135–159, https://doi.org/10.1007/s10546-017-0236-3, 2017.

Berhanu, T. A., Satar, E., Schanda, R., Nyfeler, P., Moret, H., Brunner, D., Oney, B., and Leuenberger, M.: Measurements of greenhouse gases at Beromünster tall-tower station in Switzerland, Atmospheric Measurement Techniques, 9, 2603–2614, https://doi.org/10.5194/amt-9-2603-2016, 2016.




| | data pts. | CO 90perc. | $CO_2$ 90perc. | $CH_4$ 90perc. | $H_2O$ 90perc. | $CO_2$ flx 90perc. | $H_2O$ flx 90perc. | CO 10perc. | $CO_2$ 10perc. | $CH_4$ 10perc. | $H_2O$ 10perc. | $CO_2$ flx 10perc. | $H_2O$ flx 10perc. |
|---|---|---|---|---|---|---|---|---|---|---|---|---|---|
| alldata | 843 | 8.1 | 10.3 | 7.7 | 9.4 | 8.1 | 8.8 | 14.0 | 8.9 | 16.1 | 10.3 | 8.9 | 15.3 |
| DJF | 249 | 5.2 | 4.4 | 3.6 | 7.2 | 8.4 | 9.2 | 11.6 | 9.6 | 12.0 | 9.6 | 11.2 | 14.9 |
| JJA | 172 | 4.7 | 15.7 | 4.1 | 9.3 | 12.2 | 5.8 | 20.9 | 3.5 | 34.3 | 9.3 | 7.0 | 17.4 |
| MAM | 129 | 10.1 | 8.5 | 6.2 | 3.9 | 7.0 | 13.2 | 11.6 | 8.5 | 8.5 | 17.1 | 5.4 | 19.4 |
| SON | 293 | 11.6 | 13.0 | 14.0 | 13.7 | 5.8 | 8.2 | 13.0 | 11.6 | 12.3 | 8.5 | 9.6 | 12.6 |
| day | 119 | 5.0 | 10.1 | 5.0 | 2.5 | 11.8 | 2.5 | 21.0 | 10.9 | 23.5 | 11.8 | 8.4 | 20.2 |
| night | 131 | 11.5 | 4.6 | 9.9 | 9.9 | 3.1 | 9.9 | 6.9 | 13.0 | 6.9 | 8.4 | 9.2 | 9.9 |
| morning | 84 | 8.3 | 9.5 | 3.6 | 3.6 | 10.7 | 10.7 | 8.3 | 3.6 | 20.2 | 8.3 | 4.8 | 13.1 |
| evening | 233 | 6.0 | 17.6 | 6.0 | 12.0 | 9.0 | 10.7 | 22.3 | 10.3 | 21.9 | 9.9 | 8.2 | 18.5 |
| WindNE | 197 | 16.2 | 15.2 | 15.2 | 11.7 | 6.1 | 6.6 | 9.6 | 4.6 | 10.7 | 9.1 | 7.6 | 12.7 |
| WindSE | 218 | 6.4 | 7.8 | 7.8 | 5.0 | 8.7 | 8.3 | 16.5 | 10.6 | 20.6 | 15.6 | 7.8 | 20.6 |
| WindSW | 390 | 5.6 | 7.9 | 4.6 | 10.0 | 8.7 | 10.3 | 14.9 | 10.8 | 16.9 | 9.0 | 10.5 | 13.1 |
| WindNW | 38 | 0.0 | 23.7 | 0.0 | 15.8 | 7.9 | 7.9 | 13.2 | 2.6 | 10.5 | 0.0 | 5.3 | 21.1 |

**Figure A5.** same as Fig. 5 but the upper right quadrant in Fig. 3; Sec. 2.3.2). Intersecting subsets of season and wind not shown due to small sample size.

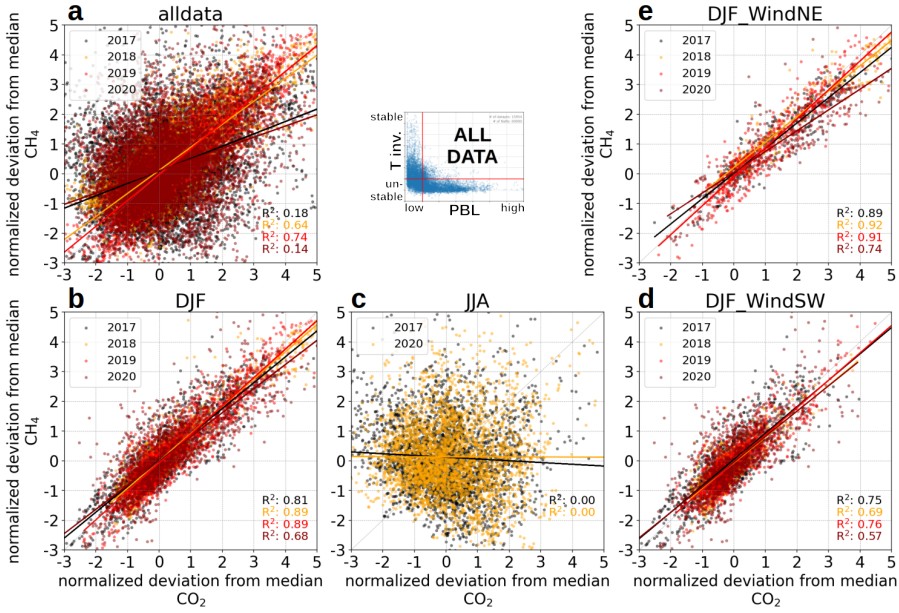

**Figure A6.** same as Fig. 7 but for $CO_2$ versus $CH_4$.

Berhanu, T. A., Szidat, S., Brunner, D., Satar, E., Schanda, R., Nyfeler, P., Battaglia, M., Steinbacher, M., Hammer, S., and Leuenberger, M.: Estimation of the fossil fuel component in atmospheric $CO_2$ based on radiocarbon measurements at the Beromünster tall tower, Switzerland, Atmospheric Chemistry and Physics, 17, 10 753–10 766, https://doi.org/10.5194/acp-17-10753-2017, 2017.




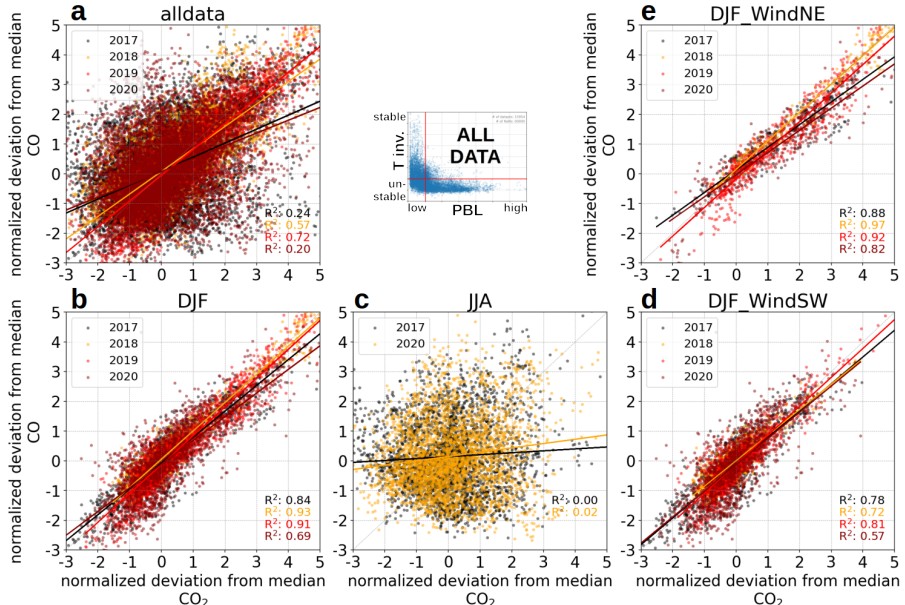

**Figure A7.** same as Fig. A6 but for $CO_2$ versus CO.

Berhanu, T. A., Hoffnagle, J., Rella, C., Kimhak, D., Nyfeler, P., and Leuenberger, M.: High-precision atmospheric oxygen measurement comparisons between a newly built CRDS analyzer and existing measurement techniques, Atmospheric Measurement Techniques, 12, 6803–6826, https://doi.org/10.5194/amt-12-6803-2019, 2019.

Crosson, E.: A cavity ring-down analyzer for measuring atmospheric levels of methane, carbon dioxide, and water vapor, Applied Physics B, 92, 403–408, https://doi.org/10.1007/s00340-008-3135-y, 2008.

Heiskanen, J., Brümmer, C., Buchmann, N., Calfapietra, C., Chen, H., Gielen, B., Gkritzalis, T., Hammer, S., Hartman, S., Herbst, M., Janssens, I. A., Jordan, A., Juurola, E., Karstens, U., Kasurinen, V., Kruijt, B., Lankreijer, H., Levin, I., Linderson, M.-L., Loustau, D., Merbold, L., Myhre, C. L., Papale, D., Pavelka, M., Pilegaard, K., Ramonet, M., Rebmann, C., Rinne, J., Rivier, L., Saltikoff, E., Sanders,

R., Steinbacher, M., Steinhoff, T., Watson, A., Vermeulen, A. T., Vesala, T., Vítková, G., and Kutsch, W.: The Integrated Carbon Observation System in Europe, Bulletin of the American Meteorological Society, 103, E855–E872, https://doi.org/10.1175/BAMS-D-19-0364.1, publisher: American Meteorological Society Section: Bulletin of the American Meteorological Society, 2022.

Henne, S., Brunner, D., Oney, B., Leuenberger, M., Eugster, W., Bamberger, I., Meinhardt, F., Steinbacher, M., and Emmenegger, L.: Validation of the Swiss methane emission inventory by atmospheric observations and inverse modelling, Atmospheric Chemistry and Physics,

16, 3683–3710, https://doi.org/10.5194/acp-16-3683-2016, 2016.

Herrmann, L.: Eddy covariance flux analysis - Notes on determination of $CO_2$ and $H_2O$ fluxes at Beromünster radio tower with EddyPro eddy covariance software, Tech. rep., Climate and Environmental Physics, Physics Institute, University of Bern, 2019.

Hiller, R. V., Bretscher, D., DelSontro, T., Diem, T., Eugster, W., Henneberger, R., Hobi, S., Hodson, E., Imer, D., Kreuzer, M., Künzle, T., Merbold, L., Niklaus, P. A., Rihm, B., Schellenberger, A., Schroth, M. H., Schubert, C. J., Siegrist, H., Stieger, J., Buchmann, N., and



Brunner, D.: Anthropogenic and natural methane fluxes in Switzerland synthesized within a spatially explicit inventory, Biogeosciences, 11, 1941–1959, https://doi.org/10.5194/bg-11-1941-2014, 2014.

Kljun, N., Calanca, P., Rotach, M. W., and Schmid, H. P.: A simple two-dimensional parameterisation for Flux Footprint Prediction (FFP), Geoscientific Model Development, 8, 3695–3713, https://doi.org/10.5194/gmd-8-3695-2015, 2015.

Kotthaus, S. and Grimmond, C. S. B.: Atmospheric boundary-layer characteristics from ceilometer measurements. Part 1: A new method to
track mixed layer height and classify clouds, Quarterly Journal of the Royal Meteorological Society, 144, 1525–1538, https://doi.org/10.1002/qj.3299, 2018.

Kotthaus, S., Haeffelin, M., Drouin, M.-A., Dupont, J.-C., Grimmond, S., Haefele, A., Hervo, M., Poltera, Y., and Wiegner, M.: Tailored Algorithms for the Detection of the Atmospheric Boundary Layer Height from Common Automatic Lidars and Ceilometers (ALC), Remote Sensing, 12, 3259, https://doi.org/10.3390/rs12193259, 2020.

Kotthaus, S., Bravo-Aranda, J. A., Collaud Coen, M., Guerrero-Rascado, J. L., Costa, M. J., Cimini, D., O'Connor, E. J., Hervo, M., Alados-Arboledas, L., Jiménez-Portaz, M., Mona, L., Ruffieux, D., Illingworth, A., and Haeffelin, M.: Atmospheric boundary layer height from ground-based remote sensing: a review of capabilities and limitations, Atmospheric Measurement Techniques Discussions, pp. 1–88, https://doi.org/10.5194/amt-2022-14, publisher: Copernicus GmbH, 2022.

Leclerc, M. Y. and Thurtell, G. W.: Footprint prediction of scalar fluxes using a Markovian analysis, Boundary-Layer Meteorology, 52,
247–258, https://doi.org/10.1007/BF00122089, 1990.

Mauder, M., Foken, T., Aubinet, M., and Ibrom, A.: Eddy-Covariance Measurements, Springer Handbook of Atmospheric Measurements, chap. 55, Springer International Publishing, https://doi.org/10.1007/978-3-030-52171-4, 2021.

Oney, B., Henne, S., Gruber, N., Leuenberger, M., Bamberger, I., Eugster, W., and Brunner, D.: The CarboCount CH sites: characterization of a dense greenhouse gas observation network, Atmospheric Chemistry and Physics, 15, 11 147–11 164, https://doi.org/10.5194/
acp-15-11147-2015, 2015.

Oney, B., Gruber, N., Henne, S., Leuenberger, M., and Brunner, D.: A CO-based method to determine the regional biospheric signal in atmospheric $CO_2$, Tellus B: Chemical and Physical Meteorology, 69, 1353 388, https://doi.org/10.1080/16000889.2017.1353388, 2017.

Pisso, I., Sollum, E., Grythe, H., Kristiansen, N. I., Cassiani, M., Eckhardt, S., Arnold, D., Morton, D., Thompson, R. L., Groot Zwaaftink, C. D., Evangeliou, N., Sodemann, H., Haimberger, L., Henne, S., Brunner, D., Burkhart, J. F., Fouilloux, A., Brioude, J., Philipp, A.,
Seibert, P., and Stohl, A.: The Lagrangian particle dispersion model FLEXPART version 10.4, Geoscientific Model Development, 12, 4955–4997, https://doi.org/10.5194/gmd-12-4955-2019, 2019.

Rust, D., Katharopoulos, I., Vollmer, M. K., Henne, S., O'Doherty, S., Say, D., Emmenegger, L., Zenobi, R., and Reimann, S.: Swiss halocarbon emissions for 2019 to 2020 assessed from regional atmospheric observations, Atmospheric Chemistry and Physics, 22, 2447–2466, https://doi.org/10.5194/acp-22-2447-2022, publisher: Copernicus GmbH, 2022.

Satar, E., Berhanu, T. A., Brunner, D., Henne, S., and Leuenberger, M.: Continuous $CO_2$/$CH_4$/CO measurements (2012–2014) at Beromünster tall tower station in Switzerland, Biogeosciences, 13, 2623–2635, https://doi.org/10.5194/bg-13-2623-2016, 2016.

Schmutz, M., Vogt, R., Feigenwinter, C., and Parlow, E.: Ten years of eddy covariance measurements in Basel, Switzerland: Seasonal and interannual variabilities of urban $CO_2$ mole fraction and flux: TEN YEARS OF $CO_2$ MEASUREMENTS IN BASEL, Journal of Geophysical Research: Atmospheres, 121, 8649–8667, https://doi.org/10.1002/2016JD025063, 2016.

Stohl, A., Forster, C., Frank, A., Seibert, P., and Wotawa, G.: Technical note: The Lagrangian particle dispersion model FLEXPART version 6.2, Atmos. Chem. Phys., p. 14, 2005.



Vermeulen, A., Schmidt, M., Ramonet, M., Messager, C., Jourd'Heuil, L., Manning, A., Gloor, M., Jordan, A., Popa, E., Thompson, R., Kozlova, E., Moors, E., Elbers, J., Jans, W., ter Maat, H., Moncrieff, J., Conen, F., Haszpra, L., Barca, Z., Szilagyi, I., Stefani, P., Miglietta, F., and Lindroth, A.: CHIOTTO Final report. Covering the period of 1 Nov. 2002 – 1 May 2006., Tech. Rep. ECN-E–07-052, Energy research Centre of the Netherlands, https://publicaties.ecn.nl/ECN-E--07-052, 2008.

Vesala, T., Kljun, N., Rannik, U., Rinne, J., Sogachev, A., Markkanen, T., Sabelfeld, K., Foken, T., and Leclerc, M.: Flux and concentration footprint modelling: State of the art, Environmental Pollution, 152, 653–666, https://doi.org/10.1016/j.envpol.2007.06.070, 2008.

WMO: 20$^{th}$ WMO/IAEA Meeting on Carbon Dioxide, Other Greenhouse Gases and Related Measurement Techniques (GGMT-2019), Tech. Rep. GAW Report No. 255, World Meteorological Organization. Global Atmosphere Watch., https://library.wmo.int/index.php?lvl= notice_display&id=21758, 2020.

Wutzler, T., Lucas-Moffat, A., Migliavacca, M., Knauer, J., Sickel, K., Šigut, L., Menzer, O., and Reichstein, M.: Basic and extensible post-processing of eddy covariance flux data with REddyProc, Biogeosciences, 15, 5015–5030, https://doi.org/10.5194/bg-15-5015-2018, 2018.