# Peer review of "Analysis of high gas concentration and flux measurements at Swiss Beromünster tall tower"

_EGUsphere, 2022_

## Referee Comment (RC2)

The study **"Analysis of high gas concentration and flux measurements at Swiss Beromunster tall tower" by Plach et al.** presents analysis on spatial and temporal variability of the greenhouse gases (GHGs) concentration and their fluxes at a tall tower 212 m above the ground level a.g.l. Measurements of GHGs at tall towers are very important since are used on inversion modelling to derive the emissions and verify the national GHG inventories created using bottom-up approaches.

The results are well presented (specially schematics), the idea is very good, however, my concern is on the approach of main method that all data are classified – transport scale. My comments are below:

**Main weakness of the study – classification of the transport scale:** the whole measurements, concentrations of $CO_2$, $CH_4$, $CO$ and $H_2O$ and the $CO_2$ and $H_2O$ flux, taken from only the highest inlet 212 m a.g.l. are classified based on the atmospheric boundary layer height (which is relative to the site where tower is located) and the potential temperature gradient. The first concern is how accurate the automatic lidars and ceilometers located 100 km away (in complex terrain) represent the local boundary conditions of site where tower is located? Regarding, the calculated potential temperature gradient is not mentioned from which heights the temperature measurements are taken for this calculation. Are there wind speed measurement available at tower? – as the atmospheric mixing is driven by the vertical wind-speed gradients as well.

The method to classify the local, distant, and regional scale has weaknesses:

1. **Local → ABL height ≤ 520 m a.g.l (tower height 212 m a.g.l.) and $dT/dz \geq 10$ K km$^{-1}$**
   I suspect that you are measuring the **local emission** if the **ABL is lower** than **concentrations/flux obtained from inlet at 212 m a.g.l.** – if the ABL would be exactly at 520 m a.g.l. then YES but if it is under inlet height than measurements would be from the **free troposphere**. If you would include the concentrations from lower inlet heights (e.g. 12 and 45 m a.g.l.) for the analysis than it would make more sense but still with caution always looking what is the ABL height.
2. **Distant → ABL height ≥ 520 m a.g.l (tower height 212 m a.g.l.) and -10 K km$^{-1}$ ≤ $dT/dz$ ≥ 10 K km$^{-1}$**
   In this group are classified the early morning / late evening measurements, mostly called as 'transitional time' when there is happening the formation/breakup of the nocturnal inversion layer or in the times of turbulences were as a contribution to the concentrations will be also the concentrations from the previous days (from the residual layer) and not sure in that case if it is distant source contribution - this is more something as transitional times where the mixing/suppression beginnings.
3. **Regional →** ABL height ≥ 520 m a.g.l (tower height 212 m a.g.l.) and $dT/dz \leq 10$ K km$^{-1}$. I would recommend that $dT/dz$ to be at least less than 0 K km$^{-1}$ if the wind speed gradients are not included.

Once the method for classifying the transport scale is revised and resubmitted, I will be happy to look again into the manuscript for the revision. The results section is not investigated into detail as to me would not make much more sense as the main method needs revision.

I have also added few technical comments:

**Technical comments**

**Introduction:** the method section (each section 2.1, 2.2, 2.2.1 and 2.3) is already introduced in the introduction in quite a lot of details and then repeated in method section. I would suggest rearranging the introduction section and mostly focus on describing the problem that you are interested in, why that is a problem and how you are going to solve that problem therefore narrowing it to introduce your aim of this research.

**Line 82 and 84:** the abbreviations FFP and CFP for Flux Footprint and Concentration footprint respectively are introduced but then again are introduced on line 101. **Line 98-99** Same is the eddy covariance (EC) and the ABL…is introduced in introduction and repeated in method section.

**Section 2.2.1:** it is described how the GHG concentrations are measured; it says that the ambient air from each height is sampled for 3 min following with the 1 min which is used to avoid contributions from the previous level. How many measurements within an hour are made in each height? 3 / 4 measurements? Are these measurements spread in that way that would represent correct way the atmospheric state for that particular hour. Could you give more technical details on this.

**Section 2.2.3:** The automatic lidars and ceilometers are located 100 km away from the tower. **Line 154** states that despite the distance the two sites are comparable climates and ABL characteristics – is there any study on this, can you show an example. Usually, in complex terrain the local conditions (formation of ABL) are quite unique for each site and especially under stable conditions.

---

## Author Comment (AC1)

**We would like to very much thank the anonymous reviewer 1 and Dafina Kikaj, reviewer 2, for reviewing our study and her/his constructive comments. Please find below the referee's comments in black font and** the authors' response in blue font.
* * *
**reviewer 1**
* * *
*General comments*
The submitted study presents an analysis of tall-tower concentration and flux measurements in Switzerland. Such towers measuring both trace gas molar fractions and fluxes for several years are quite unique and could indeed provide important additional data to improve regional inversions aiming to constrain regional and national level emissions/removals of GHGs.

The authors furthermore implement an impressive, and in my view defensible, statistical approach to identify "high" concentration and flux values, taking into account seasonal and diurnal variations. After identifying the seasonal-diurnal thresholds for high values, an analysis was conducted explain test whether local, regional or distant sources were responsible for values exceeding 90th percentiles, by splitting the dataset into local, regional and distant subsets, by inferring transport distances from the interplay between mixing height (relative to tower height) and vertical potential temperature gradient. If the subset contains more than 10% of values classified as high, the authors inferred that significant trace gas sources are found within this transport scale.

The paper is well-written and results are well presented; however, I have substantial concerns over the methods used to classify the transport scales and whether such a common approach can be applied to both concentrations and fluxes. Furthermore, while the writing is good and the graphs are sharp and well rendered, I was disappointed that the study was not put into context were a more thorough review of literature in the introduction and then a discussion of what this tower and these results mean within this context. I elaborate on these and other major concerns in the section Specific comments.

We agree that the introduction and the discussion in the current version of the manuscript is too much focused on the Beromünster site, and we will acknowledge this by broadening the introduction (e.g. referring to Dang et al, 2019; especially Fig. 1 therein, and Lelandais et al., 2022) and discussion section in the revised version.

Our classification of source/transport scale into LOCAL, DISTANT, and REGIONAL sources is a necessary simplification. We agree that each of these three classifications is a mixture of transport scales to some degree. However, the fact remains that we see distinct differences in the number of high concentrations/fluxes when analysing the data separated by ABL height/potential temperature gradient. We will elaborate further below.

*Specific Comments*
**Characterization of transport scales**. This constitutes the biggest weakness of the study, which is critical given that it is fundamental to the goals of the analysis. ABL height above the tower height coupled with moderately stable to unstable stratification (Pot T gradient > - 10 K km-1) was considered Regional. Relative to a Distant scale, I can imagine this would be valid for concentrations but not for fluxes. I think within the range

of ABL heights and stratification considered here as regional, one would actually have flux footprint ranging over 2 or more orders of magnitude (shorter footprints with high ABLs and/or very unstable conditions; longer footprints with ABLs just above the tower height and/or stable stratification).

We agree that it is challenging to use the same classification for concentration and fluxes and we are aware of the limitations and will further discuss these limitations in the revised version and also further emphasise the difference of our classification for concentrations and fluxes.
As an example, the flux footprint (FFP) parameterization is only valid for situations with ABL heights above the tower, otherwise the fluxes become decoupled from surface fluxes and this parameterization can not be used to estimate the FPPs anymore. However, fluxes are still measured in the "ABL below tower" cases. With this limitation an association of fluxes and corresponding sources becomes more challenging. Unfortunately, a classification of fluxes in all transport scales following FFPs is therefore not possible. Obviously FFPs are of a much smaller scale than concentration footprints (CFPs). However, we do not claim to be able to give exact distances to the sources. Furthermore, our classification shows much less distinct differences for the fluxes. In lack of a better alternative, we therefore think it is reasonable to keep our source/transport scale classification for concentrations and fluxes. However, we will emphasise the differences and the particular challenges in our classification concerning fluxes. Also see our summarising statement at the end of this document.

Furthermore, for the concentrations, I believe it is not possible to then split the remaining conditions when ABL < Tower height between local and distant scales. I think with ABL below the tower, the scale is distant. In this study, an ABL < tower height coupled with a very strong inversion is considered local. I think here the assumption is that the ABL is in fact below the tower base, so that measurements are only sensitive to sinks/sources on the elevated local terrain around the tower. I can imagine under such a scenario, the concentration observations are proportionally more sensitive to local sources than regional ones trapped under the inversion; however, surely this scenario would constitute a local + distant scenario. Moreover, I think the ABL height being below the tower base would be decisive rather than the Pot T gradient at the tower.

Yes, it is our assumption that local sources are dominating if the ABL is below the tower base. And again, we acknowledge that the separation between DISTANT and LOCAL sources is a simplification and each classification set is a mixture of transport scales. However, we argue that DISTANT and LOCAL sources dominate in the respective sets. Furthermore, we disagree with the last sentence, stating that the ABL would be decisive rather than the potential T gradient. If this were true, we would not see any differences in the percentages of high concentrations between our LOCAL and DISTANT cases. However, there are differences, as can be seen in Fig. 5 in the preprint (e.g., comparing the WindNE sets of LOCAL (bottom table) and DISTANT (top table). Overall our DISTANT set shows higher observations, especially in the WindNE set.

Finally, I think the applied scheme ignores important factors affecting fluxes in that low ABL/stable stratification leads to decoupling between the turbulent fluxes and the sources/sinks in the footprint and an associated increase in storage and advection fluxes below receptor height. For example, under the distant subset, one could potentially see more high fluxes at midday not because of distant sources but because the 90th percentile

is negative flux value closer to zero, and under such conditions absolute exchange is lower because of the dominating non-turbulent fluxes within the volume below the receptor.

We are well aware that our classification is a simplification as with the two parameters, ABL and potential temperature, we cannot expect a complete separation of the mentioned transport/source cases. Therefore, our classification does always represent a certain ratio of transport cases yet with prioritisation to the respective cases used for naming the transport scale sets. In this regard, we will also add a short description of the complex physical nature of the diurnal ABL dynamics and their interpretation (e.g., referring to Dang et al., 2019; Stoll et al., 2020; Porté-Agel, 2018).
Concerning the mentioned example of high fluxes at midday because of fluxes being closer to zero, we refer to our definitions for high values (section 2.3.1 in the preprint), which should remove any diurnal and seasonal signal caused by vegetation activities by calculating statistics separately for 2-hour-windows of each month (see Fig. A1 in the preprint showing statistics for the entire data set before any separation by transport scale – seasonal sets (DJF, JJA, MAM, SON) show no difference in the number of high values; also diurnal sets (day, night, morning, evening) show no corresponding differences).

I therefore must express considerable doubts about the validity of this scheme for defining the three transport scales. If the authors were to revise and resubmit the analysis, I would strongly recommend basing this analysis on concentration and flux footprints.

Concentration footprints for the Beromünster tower are published based on the shorter time series by Oney et al., 2015 (Fig. R1). They are as expected considerably larger (complete Swiss plateau) than the flux footprints (<60 km extension). The reviewer's suggestion to base the classification of transport scale on the footprints would unfortunately not work as the FFPs for ABLs below the tower height can not be calculated, i.e. for the LOCAL and DISTANT cases (Fig. R2). This is because the FFP parameterization is not valid for the "ABL below tower" occasions as the fluxes become decoupled from the surface, as stated by the reviewer above.

[Figure]

Fig. R1: Figure 12 from Oney et al. (2015) showing concentration footprints for Beromünster (dark blue line) for (a) summer, (b) winter, (c) summer: day, (d) summer:night.

[Figure]

Fig. R2: Sketch of the diurnal evolution of the ABL. Approximate times of our four time of day selections (night, morning, day, evening) are marked with shaded areas. Modified after Dang et al. (2019). Abbreviations: residual layer (RL),stable boundary layer (SBL), early morning transition (EMT), convective boundary layer (CBL), entrainment zone (EZ), early evening transition (EET). We will include a similar sketch in the revised version (at least as a supplement).

The authors mention that follow-up work is planned to model flux and concentration footprints with FLEXPART, which I think would have been very helpful for the aims of the submitted study. Assuming that this work may take time to emerge, perhaps the authors could nonetheless simpler e.g. analytical footprint models to split the data into different transport scales. I think presentation of footprints, even if coarser approximations, of both

concentration and flux footprints would be useful to this study.

Indeed it would be helpful to have both CFPs and FFPs available from FLEXPART model calculations. Unfortunately, this is not possible in a reasonable time frame for this present study. Oney et al. (2015) show Beromünster CFPs from FLEXPART model simulations (Fig. R1) for a shorter time series. The CFPs extend to the entire Swiss plateau. We will describe their results briefly in the revised version in order to allow for a better interpretation of our results.

**Lack of context**. The Introduction focused too much on previous studies at the Beromunster tower and failed to introduce the context and rationale for the measurements and the analysis. I would have expected the introduction to touch e.g.:

- Need for national and regional networks of in situ trace gas measurements for improved inverse modelling of regional/national fluxes and emissions
  - CH (and UK) as examples of countries establishing national in situ networks for this purpose
- Particular global/regional need for tall-towers…
  - ICOS aims at certain ratio between mountain stations and tall-towers
- Novelty of Beromünster, operationally measuring both trace gas concentrations and fluxes. There are not many others... there are however, some and would be good to name a couple of these examples

After setting the scene, it would then be good to justify why this particular analysis was undertaken.

Additionally, the literature cited in the introduction and discussion was heavily dominated by previous studies of data from Beromünster, with almost no reference to relevant studies from elsewhere.

Reflecting that this information was missing/lacking, the paper therefore does not communicate what authors wanted to do with this analysis and why, and what new insights have been gained. For example, what new insights were gained from correlations between trace gases compared to other studies elsewhere and previous analysis at the tower.

We agree that the introduction in the current version of the manuscript is too much focused on the Beromünster site, and we will broaden the introduction in the revised version (e.g., referring to Dang et al, 2019; especially Fig. 1 therein, and Lelandais et al., 2022; Stoll et al., 2020; Porté-Agel, 2018) and address the points mentioned above.
We will further extend the discussion with appropriate references to other studies not focusing on Beromünster and put our results in context to these studies. We will also include a short discussion on the complex ABL dynamics (Fig. R2).
Furthermore, we will emphasise our scientific contribution with this study. Given the example of our correlation analysis, on the one hand, we confirmed the findings of a previous study (Satar et al, 2016; for a shorter time series) which only showed the correlation without discussing the processes causing the $CO_2/CH_4$ correlation. We argue that this correlation must stem from mixing processes. We will make this clearer in the revised manuscript. Moreover, we extended the correlation analysis to also include the wind direction, which we find to play an important role on the likelihood to observe high concentrations/fluxes.

**Other specific comments**
In addition to the above, I would also highlight the following issues.

- Use of gap-filled EC data. I think such an analysis should be using only measured fluxes and not measured and gap-filled. For a gap-filled data point, the data will not reflect e.g. the transport scale at the time but an average of those data points used to gap-fill
- Show the in the supplements the monthly diurnal curves for trace gas concentrations and fluxes and perhaps show the stats per hour e.g. percentiles and means
- Consider removing H2O conc/fluxes from the analysis or at least make adequate reference to these results

Gap-filling is an established procedure when studying fluxes. In order to have a sufficient number of data points, we therefore think it is reasonable to also use gap-filled data in our analysis. However, we will provide additional information on how much data points are gap-filled in the supplements of the revised version. Furthermore, we will show the monthly-diurnal curves of concentrations/fluxes used for our statistical definition of $90^{th}/10^{th}$ percentiles. Furthermore, we will consider moving the $H_2O$ conc./flux discussion to the supplement to streamline the revised manuscript. Also see summarising statement at the end of this response.

**Technical corrections**
Given fundamental scientific concerns and my recommendation that the manuscript and the analysis itself be substantially revised, I think it inappropriate to list technical issues with the manuscript. Furthermore, it is worth repeating that, despite my scientific concerns, the MS is well-written and with good graphical presentation of the results. I did not therefore come across many technical issues such as typos, spelling and grammar mistakes.
* * *
**Dafina Kikaj (reviewer 2)**
* * *
The study "**Analysis of high gas concentration and flux measurements at Swiss Beromunster tall tower" by Plach et al.** presents analysis on spatial and temporal variability of the greenhouse gases (GHGs) concentration and their fluxes at a tall tower 212 m above the ground level a.g.l. Measurements of GHGs at tall towers are very important since they are used on inversion modelling to derive the emissions and verify the national GHG inventories created using bottom-up approaches.
The results are well presented (specially schematics), the idea is very good, however, my concern is on the approach of the main method that all data are classified – transport scale. My comments are below:

**Main weakness of the study – classification of the transport scale:** the whole measurements, concentrations of CO2, CH4, CO and H2O and the CO2 and H2O flux, taken from only the highest inlet 212 m a.g.l. are classified based on the atmospheric boundary layer height (which is relative to the site where the tower is located) and the

potential temperature gradient. The first concern is how accurately the automatic lidars and ceilometers located 100 km away (in complex terrain) represent the local boundary conditions of the site where the tower is located? Regarding, the calculated potential temperature gradient is not mentioned from which heights the temperature measurements are taken for this calculation. Are there wind speed measurements available at the tower? – as the atmospheric mixing is driven by the vertical wind-speed gradients as well.

Yes, the potential temperature gradient is calculated as the difference between the lowest level of the tower (12.5 m agl) and the highest level (212.5m agl). We will clarify this in the revised version. The vertical wind speed measurements are available only at the toplevel of Beromünster tower. We will add a short analysis including vertical wind speeds in the supplements of the revised version (and show corresponding figures further below in this document).
Concerning how representative the Payerne ABL measurements are for Beromünster. Since Beromünster is located in the Swiss Plateau, a landscape with relative smooth topography (compared to the Alps further south), we argue that the Payerne ABL should be representative for Beromünster, a fact which can also be seen from the tower location on days with fog. The tower is above a smooth fog layer covering the valleys below (due to copyright concerns, we only provide a link to a corresponding google search here, e.g., https://www.google.at/search?q=berom%C3%BCnster+turm+nebel&tbm=isch&ved=2ahU KEwjCp5DY8Pn-AhWZmScCHaVPBjAQ2-cCegQIABAA&oq=berom%C3%BCnster+turm+ nebel&gs_lcp=CgNpbWcQAzoECCMQJzoECAAQHICmAljgGWDQGmgCcAB4AIABd4gB ywSSAQMzLjOYAQCgAQGqAQtnd3Mtd2l6LWltZ8ABAQ&sclient=img&ei=iXxjZIK_Cpmzn sEPpZ-ZgAM&bih=763&biw=1597#imgrc=0aA-Fd9WbqVfRM )

The method to classify the local, distant, and regional scale has weaknesses:

**1. Local → ABL height ≤ 520 m a.g.l (tower height 212 m a.g.l.) and dT/dz ≥ 10 K km -1**
I suspect that you are measuring the local emission if the ABL is lower than concentrations/flux obtained from inlet at 212 m a.g.l. – if the ABL would be exactly at 520 m a.g.l. then YES but if it is under inlet height than measurements would be from the free troposphere. If you would include the concentrations from lower inlet heights (e.g. 12 and 45 m a.g.l.) for the analysis than it would make more sense but still with caution always looking what is the ABL height.

We are aware that our classification in LOCAL, DISTANT, and REGIONAL is a simplification. However, as stated by reviewer 1, our assumption is that LOCAL also includes situations where the ABL height is below the tower base, so local sources in the immediate vicinity of the tower should be the dominant (or at least a strongly contributing) source.

**2. Distant → ABL height ≥ 520 m a.g.l (tower height 212 m a.g.l.) and -10 K km-1 ≤ dT/dz ≥ 10 K km-1**
In this group are classified the early morning / late evening measurements, mostly called as 'transitional time' when there is happening the formation/breakup of the nocturnal inversion layer or in the times of turbulences were as a contribution to the concentrations will be also the concentrations from the previous days (from the residual layer) and not sure in that case if it is distant source contribution - this is more something as transitional times where the mixing/suppression beginnings.

Fig. R3 shows time of observation histograms falling into our transport scale sets. As you can see, there is a distinct low number of observations in the afternoon in the DISTANT set, however there is an almost equal distributed number of observations between 20:00 and 08:00. This histogram shows that our data not just includes the transition times mentioned above.

[Figure]

Fig. R3: Top row: Data points in the ABL vs. local potential T gradient space for the respective data selections (LOCAL, DISTANT, REGIONAL, No name). Bottom row: Histograms of the corresponding times of observations (this figure will also be added as a supplement to the revised manuscript).

**3. Regional → ABL height ≥ 520 m a.g.l (tower height 212 m a.g.l.) and dT/dz ≤ 10 K km -1.**
I would recommend that dT/dz to be at least less than 0 K km-1 if the wind speed gradients are not included.

Using 0 K km$^{-1}$ as a threshold between our classifications would result in almost no data points in the DISTANT and REGIONAL transport scale selections as can be seen in Fig. R3 (top row).

Once the method for classifying the transport scale is revised and resubmitted, I will be happy to look again into the manuscript for the revision. The results section is not investigated into detail as to me would not make much more sense as the main method needs revision.

Figure R4, R5, and R6 show the analysed data points as functions of ABL height, pot. T gradient, and additionally vertical wind speed. In contrast to the data shown in the preprint, we skip data points below 260m agl due to the high uncertainty level associated with the ABL measurements in this low heights (something we all consider to do for the revised manuscript; the low values are mostly associated with DJF and SON – Fig. R7; and especially day and morning – Fig. R8). Vertical wind speed measurements at the top level of the Beromünster tower indicate that our selection scheme of LOCAL, REGIONAL and DISTANT is valid as the LOCAL set corresponds to lower mean vertical speed for higher pot. T gradients compared to the REGIONAL and DISTANT set (Fig. R5). Furthermore, the

vertical wind speeds of the DISTANT set ranges to higher values compared to the LOCAL set (Fig. R6). Yet again, it is obvious that all selected transport scale sets represent mixtures of different transport scale cases. However, we argue that our classification is following the dominant transport scale of the respective sets.

[Figure]

Fig. R4: Analysed data points in the ABL vs. pot. T gradient space; same as Fig. 3 in the preprint. Transport scale selections are indicated by colours: (yellow) LOCAL - low ABL, (red) DISTANT, (orange) REGIONAL, (blue) no name. Values represent 6h running means (RM) to reduce noise. Fig. R5 and R6 use the same colour scheme.

[Figure]

Fig. R5: Analysed data points shown in the pot. T gradient vs. vertical wind speed space; same colour scheme as in Fig. R4: (yellow) LOCAL - low ABL, (red) DISTANT, (orange) REGIONAL, (blue) no name. Values represent 6h running means (RM) to reduce noise.

[Figure]

Fig. R6: Analysed data points shown in the ABL height vs. vertical wind speed space; same colour scheme as in Fig. 4: (yellow) LOCAL, (red) DISTANT, (orange) REGIONAL, (blue) no name. Values represent 6h running means (RM) to reduce noise.

I have also added few technical comments:

**Technical comments**

**Introduction:** the method section (each section 2.1, 2.2, 2.2.1 and 2.3) is already introduced in the introduction in quite a lot of details and then repeated in method section. I would suggest rearranging the introduction section and mostly focus on describing the problem that you are interested in, why that is a problem and how you are going to solve that problem therefore narrowing it to introduce your aim of this research.

We agree, the introduction will be revised to put Beromünster in a broader context including other (similar) sites. As you suggest, we will also revise in order to avoid repetitions of the methods.

**Line 82 and 84:** the abbreviations FFP and CFP for Flux Footprint and Concentration footprint respectively are introduced but then again are introduced on line 101. Line 98-99 Same is the eddy covariance (EC) and the ABL...is introduced in introduction and repeated in method section.

Thank you, will be revised accordingly.

**Section 2.2.1:** it is described how the GHG concentrations are measured; it says that the ambient air from each height is sampled for 3 min following with the 1 min which is used to avoid contributions from the previous level. How many measurements within an hour are made in each height? 3 / 4 measurements? Are these measurements spread in that way that would represent correct way the atmospheric state for that particular hour. Could you give more technical details on this.

The measurement time for each height level is three minutes of which the first minute is skipped to avoid mixing effects from the previous level. This corresponds to four measurements per hour and height level.

**Section 2.2.3:** The automatic lidars and ceilometers are located 100 km away from the tower. **Line 154** states that despite the distance the two sites are comparable climates and ABL characteristics – is there any study on this, can you show an example. Usually, in complex terrain the local conditions (formation of ABL) are quite unique for each site and especially under stable conditions.

Unfortunately, we are not aware of any studies looking at the ABl dynamics at our site. However, we argue, as previously stated, that the terrain around Beromünster is actually not that complex, at least compared to the Alps further south.
* * *
**Summarising statement**

We acknowledge the concerns of both reviewers with our source/transport scale classification and we want to emphasise once more that we are aware of the fact that any classification like this will always be a mixture of cases with different transport scales. However, we think our classification describes the dominant number of cases in the respective classification sets. Unfortunately, the suggested changes to the classification are not feasible as argued above - (1) using concentration and flux footprints as a basis for the classification – the FFP parameterization is not valid for "ABL below tower", (2) adjusting the pot. T gradient cutoff to 0 K km$^{-1}$ – hardly any measurements in DISTANT and REGIONAL transport scale sets (see Fig. R3). Furthermore, changing the classifications would still result in mixtures of different transport scale cases to some degree. In fact, we don't think it is possible to define a classification resulting in an exact separation between transport scales.

However, we will address the reviewer's concerns in the revised version and sharpen our classification and discussion correspondingly. We will especially focus on describing the differences of our classification for concentrations and fluxes. Obviously, FFPs are much smaller than CFPs. We do not claim to be able to provide exact transport distances with our classification, however we see differences in the number of high concentrations/fluxes in the respective transport scale/source sets. Although, for the fluxes our analysis shows much less pronounced differences.

Furthermore, we are also considering to streamline the manuscript by:
- moving the analysis of H$_2$O concentrations and fluxes to the supplement since the results are much less distinct as with the other species.
- moving seasonal (SON, MAM) and diurnal (morning, evening) transition periods to the supplements in the revised version to be able to focus more on the distinct differences between the diurnal and seasonal sets (DJF vs. JJA, and day vs. night, the latter indicated in Fig. R2).

**Thank you again, for all your time and effort in acting as reviewer and editor, respectively!**

**Additional figures**

[Figure]

Fig. R7: Same as Fig. 8 in the preprint ("alldata" is the same), but for all four seasons. Note, most ABL heights near 0 m agl are observed in winter (DJF) and fall (SON).

[Figure]

Fig. R8: Same as Fig. 8 in the preprint ("alldata" is the same), but for all four time of day selections. Note, most ABL heights near 0 m agl are observed during night and morning.

[Figure]

Fig. R9: Same as Fig. R7, but only including times of day, which are included in the daytime selections (see Fig. R8), i.e., observations during times indicated in white in Fig. R2 are removed. Note, most ABL heights near 0 m agl are observed in winter (DJF) and fall (SON).

**References**

**Dang et al. (2019):** A Review of Techniques for Diagnosing the Atmospheric Boundary Layer Height (ABLH) Using Aerosol Lidar Data. Remote Sens. 2019, 11(13), 1590; https://doi.org/10.3390/rs11131590

**Lelandais et al. (2022):** Analysis of 5.5 years of atmospheric CO2, CH4, CO continuous observations (2014–2020) and their correlations, at the Observatoire de Haute Provence, a station of the ICOS-France national greenhouse gases observation network. Atmospheric Environment, 277, 15 May 2022, 119020; https://doi.org/10.1016/j.atmosenv.2022.119020

**Oney et al. (2015):** The CarboCount CH sites: characterization of a dense greenhouse gas observation network. ACP, 15, 11147–11164, 2015; https://doi.org/10.5194/acp-15-11147-2015

**Porté-Agel et al. (2020):** Wind-Turbine and Wind-Farm Flows: A Review. Boundary-Layer Meteorology (2020) 174:1–59; https://doi.org/10.1007/s10546-019-00473-0

**Satar et al. (2016):** Continuous $CO_2/CH_4/CO$ measurements (2012–2014) at Beromünster tall tower station in Switzerland. Biogeosciences, 13, 2623–2635, 2016; https://doi.org/10.5194/bg-13-2623-2016

**Stoll et al. (2020):** Large-Eddy Simulation of the Atmospheric Boundary Layer. Boundary-Layer Meteorology (2020) 177:541–581; https://doi.org/10.1007/s10546-020-00556-3